# Isomers Recognition in HPLC-MS/MS Analysis of Human Plasma Samples by Using an Ion Trap Supported by a Linear Equations-Based Algorithm

**DOI:** 10.3390/ijms241311155

**Published:** 2023-07-06

**Authors:** Marco Pallecchi, Luigi Lucio, Laura Braconi, Marta Menicatti, Silvia Dei, Elisabetta Teodori, Gianluca Bartolucci

**Affiliations:** Neurofarba Department, Section of Pharmaceutical and Nutraceutical Sciences, University of Florence, Polo Scientifico, Via U. Schiff 6, Sesto Fiorentino, 50019 Firenze, Italy; marco.pallecchi@unifi.it (M.P.); luigi.lucio@stud.unifi.it (L.L.); laura.braconi@unifi.it (L.B.); marta.menicatti@gmail.com (M.M.); silvia.dei@unifi.it (S.D.); elisabetta.teodori@unifi.it (E.T.)

**Keywords:** ion trap, ERMS, breakdown curves, collision energy, enzymatic degradation kinetics

## Abstract

The tandem mass spectrometry (MS/MS) approach employing an ion trap mass analyzer (IT) was evaluated in isomers recognition. The proposed approach consists of sole, simple, and rapid liquid chromatographic separation (HPLC) without requiring resolution between the analytes. Then, the MS/MS properties were optimized to solve the signal assignment using post-processing data elaboration (LEDA). The IT-MS/MS experiment uses the same site, helium as collision gas, and different time steps to modify the applied conditions on the studied ions. Nevertheless, helium cannot ensure the quick energization of the precursor ion due to its small cross-section. Then, different combinations between excitation amplitude (ExA) and excitation time (ExT) were tested to achieve the activation of the fragmentation channels and the formation of the MS/MS spectrum. Usually, the IT-MS/MS acquisition cycle is longer for other multistage instruments, decreasing the frequency of sample data collection and influencing the chromatographic profile. To solve these problems, two time segments were set up, and the elution conditions were optimized with a compromise between peaks distinction and run time reduction. The developed HPLC-MS/MS method was checked and applied to analyze a series of human plasma samples spiked with an equimolar mixture of pair of isomers.

## 1. Introduction

Mass spectrometry (MS) is an analytical technique widely used in many scientific fields to obtain quali-quantitative information on the studied compounds in different sample matrices [1,2]. Indeed, MS is an attractive technology as it combines reliable signal features, such as selectivity, detection sensitivity, wide dynamic range, and high-throughput capabilities [3]. For these reasons, the number of MS applications is increasing in analytical determinations to support most chemical investigations (environmental protection, food safety, pharmaceutical, etc.) [4,5]. Unfortunately, the specificity required for compound distinction has always been a challenge for MS users in isomer recognition. In fact, these compounds are characterized by the same elemental composition, equal molecular weight, and, often, common fragment ions. However, many strategies have been developed to overcome the problem, and some of these involve the exploration of the features of tandem mass spectrometry (MS/MS) experiments by allowing the characterization and quantification of isomers and/or isobars in mixtures via a standardized approach, applicable to different compounds [6,7,8,9,10,11,12,13,14,15,16]. With regard to this topic, our group proposed and developed LEDA (Linear Equation of Deconvolution Analysis), a mathematical algorithm that allows the distinction of isomer compounds via the elaboration of MS/MS data without their chromatographic separation [17,18,19,20]. Profiting from the advantages offered by the LEDA tool, we introduced a basic liquid chromatography (HPLC) approach that greatly simplifies the operating parameters, using a short column and a fast elution gradient. Following this proposal, the chromatographic column was used only to avoid or limit the interference of the sample matrix towards the analytes ionization process (matrix effects), leading to higher productivity without losing the specificity of their determination [21]. Then, the same HPLC conditions can be used with a large number of analytes without worrying about their separation, and the LEDA elaboration of the MS/MS data assures the signal distinction of the compounds. The reliability of this procedure allowed us to carry out a kinetic study on the enzymatic hydrolysis of a pair of isomers present simultaneously in plasma samples, evaluating their different degradation profiles [22].

All reported applications were performed, developed, and evaluated using a triple quadrupole (QqQ) system with a conventional collision cell, a collision-induced dissociation (CID) mechanism, and argon as collision gas.

Is it possible to transfer the MS/MS data post-processing LEDA approach to another type of tandem mass spectrometry system?

To answer this question, we propose using an ion trap (IT) mass spectrometer to distinguish a mixture of isomers by taking advantage of the LEDA algorithm. In order to evaluate the IT-MS/MS system, we replicated the study of plasma stability of the pair of isomers named ELF94 and ELF96, as previously reported [22]. The studied compounds are positional isomers characterized by a tertiary amine group carrying two polymethylene chains of variable length (three or six carbon atoms) linked to different aromatic moieties through two ester bonds [23]. The structures of the studied compounds are reported in Figure 1.

The aim of this work is to optimize, develop and evaluate a system coupled with LEDA data elaboration in the isomers distinction by comparing the results with those obtained by a QqQ tandem mass spectrometer.

## 2. Results

The application of the LEDA approach in isomers recognition involves the MS/MS data elaboration from a multistage MS system, such as the QqQ mass spectrometer. However, IT-MS analyzers can also perform tandem MS analysis, although it manages the experiment differently. IT operates the multistage MS experiments in the same site, using the time to modify the conditions applied to studied ions [24]. Then, a sequence of time-dependent steps must be made to perform the selection of precursor ion, its fragmentation by CID mechanism, and analysis of resulting product ions (Pis) [25,26]. For this reason, the MS/MS acquisition cycle on IT can be longer with respect to other multistage instruments, decreasing the frequency of sample data collection and influencing its chromatographic profile. Nevertheless, the MS/MS analysis from IT instruments shows some advantages: the acquisition of all product ions into a defined *m*/*z* range, using a product ion scan experiment, different excitation energy management, and reiteration of the tandem MS experiment on a product ion (MS^n^) [27]. Taking into account all the information reported above, we planned a series of experiments to investigate each feature of IT and set up the most suitable MS/MS parameters to allow the isomers recognition by the LEDA approach.

The parameters evaluated in the proposed study were as follows:Energetic dimension of CID mechanism in IT-MS/MS experiments;Interpretation of MS/MS spectra;Check of the achieved chromatographic separation;Assessment of the LEDA quantitative performances;Application of the optimized conditions of the LEDA approach in plasma stability experiments.

### 2.1. Energetics Involved in the CID Mechanism by IT-MS/MS Experiments

In the IT-MS/MS experiment, the precursor ion is energized by applying an excitation amplitude (ExA) for an established time (excitation time or ExT) in order to raise the number of collisions with helium gas, normally present inside the IT. The combination of ExA and ExT allows the activation of the fragmentation channels of precursor ions and the formation of the Pis. To establish the proper values of ExA and ExT for each analyte, a series of energy-resolved mass spectrometry (ERMS) experiments were carried out. During each ERMS experiment, the ExA value is increased from 0 to 50 (arbitrary units or a.u.) by applying a fixed ExT. Then, the ERMS procedure was repeated at ExT values of 10, 25, 50, and 100 milliseconds (ms). The evaluation of these experiments was carried out by plotting the survival yield curves of the precursor ion (SY), Pis formation (PiF), and the Pis yield (PiY, see details in Section 4.4). A typical graph used to describe the CID energetic study is reported in Figure 2, while all other graphs are in the Appendix A.

The PiF curve (red line) represents the relative abundance of the product ions vs. the total ions detected (see Equation (2)). Unfortunately, this curve fails to describe the effectiveness of the CID process; in fact, the ratio of abundances used to calculate each data point neglects the signal loss due to the precursor ion ejection during the excitation process or caused by undetected product ions because their *m*/*z* are below the low-mass cut-off of the IT-MS/MS experiment. In both cases, the product ions abundance ratio is overestimated. Then, to establish a reliable value of fragmentation yield from the CID process, the sum of abundances of the product ions should be referred to as the averaged abundance of the precursor ion before its fragmentation. In this way, a curve of PiY can be plotted vs. the ExA (green dashed line) that represents the efficiency of the CID process in tested conditions.

Analyzing these graphs, it is possible to note that each PiY curve increases, reaching a maximum value (PiYmax); the related ExA (ExA_max_) represents the energy amplitude to apply in the MS/MS process, achieving the highest Pis abundances. The obtained data of PiY_max_ and ExA_max_ for the studied isomers are reported in Table 1.

Another analysis of the graphs shown above can be done by comparing the profiles of precursor SY plots. It is worth noting that each precursor curve (black line) crosses the product ions formation line (red line) at 50% value of abundance: the corresponding ExA value is characteristic for each compound that undergoes the CID process and represents the ExA required to fragment the 50% of precursor ion (SY_ExA50_) [9]. The graph, obtained by merging the SY curves from the different ExT experiments, shows that the SY_ExA50_ value decreases accordingly to the growth of ExT, following an exponential trend. Therefore, the differences between the calculated SY_ExA50_ values gradually decrease as ExT increases. This behavior demonstrates that low values of ExTs are unable to effectively transfer the excitation energy to the precursor ion, influencing its excitation and subsequent decay. On the contrary, after 50 ms, the SY curves show close profiles and similar estimated SY_ExA50_ values. The graph of merged SY curves obtained at different ExT values for the ELF94 isomer is reported in Figure 3, while the corresponding graph of ELF96 is shown in Appendix A.

The ERMS investigation to estimate the proper CID conditions for each analyte showed the best values combination of ExA at 35 a.u. and ExT at 50 ms.

### 2.2. Collision-Induced Dissociation Study

The studied isomers (see Figure 1), in positive ions electrospray ionization (ESI), showed only an abundant signal of protonated molecules ([M+H]^+^ species) with the same *m*/*z* value (Appendix A). Therefore, a tandem mass spectrometry study was carried out to select the characteristic Pis that allow the isomers recognition. To enhance the differences between the isomers in their fragmentation pathway, the ERMS experiments were performed at an ExT of 50 ms. The ERMS data of analytes and internal standard (IS) were processed, the collision breakdown curves plotted, and the results of ELF94-ELF96 were compared in Figure 4, while the collision breakdown curves of IS were reported in Appendix A. It is interesting to observe that, after the activation of fragmentation channels, the Pis composition remains constant, and their formation curves follow the same trend showing a maximum abundance at the same ExA. This CID pattern is very different compared to the related CID graph obtained with QqQ, and it demonstrates that in IT, the precursor ion fragmentation channels are simultaneously activated at a defined ExA. Hence, after the precursor ion is completely dissociated, the IT-MS/MS spectra are the same despite ExA increases; only the PiY changes, achieving the highest value at ExA_max_.

Concerning the fragmentation pathway of the analytes, the use of IT confirmed the general scheme proposed in the previous work by using a QqQ analyzer (Appendix A), with some differences in the relative abundances and composition of product ions [22]. In fact, a new fragment is present (Pi_7_) in the MS/MS spectra of both isomers at 384 *m*/*z*, to which the structures shown in Figure 5 could be assigned.

Another important difference found, compared to previous QqQ-MS/MS data, is the low abundance of the Pi_2_ at 195 *m*/*z* (≤5%). This instance could happen both for its different formation yield from the precursor ion and for its signal loss due to the low-mass cut-off effect [24]. However, the lack of abundance of Pi_2_ in the IT-MS/MS spectra of the analytes radically changes the setup of the LEDA algorithm. The main signals (abundance ≥ 20%) in the IT-MS/MS spectra are confirmed as being Pi_1_ at 221 *m*/*z*, Pi_3_ at 392 *m*/*z,* and P_4_ at 366 *m*/*z,* and their characteristic abundances should allow the isomers distinction. Finally, Pi_5_, Pi_6,_ and the new Pi_7_ are also present and, despite their low intensity (≤10%), could be useful in the LEDA setup. Indeed, all these ions show different abundances according to the origin precursor isomer; then, it should be possible to arrange a suitable “mathematical device” able to distinguish the common signals and elaborate the MS/MS data by assigning the proper abundance to the isomer or isomers present in the sample.

### 2.3. Chromatographic Separation

The recognition of the isomers with the LEDA approach from the unresolved chromatographic peak needs the elaboration of the characteristics of Pis and reference ion (Ri) signals, but to convert them into quantitative data, the IS signal must also be recorded. Since the characteristic IT tandem mass spectrometry experiment involves the acquisition of entire spectra of Pis (product ions scan mode), three different MS/MS events must be employed to ensure the monitoring of requested signals. Each MS/MS event is time-consuming; therefore, it is necessary to separate at least the IS peak from the unresolved isomers peak by chromatography. In this way, two time segments are set up; the first allows the monitoring of IS, while the second contains the MS/MS events that alternately acquire the Ri and Pis signals of the isomers. To achieve this aim, a longer HPLC column with respect to previous approaches (30 mm) was used, optimizing the elution conditions with a compromise between peaks distinction and run time reduction. The obtained chromatographic profiles showed an appropriate separation between the IS and the analytes peaks, allowing their monitoring in different time segments. However, an acceptable resolution between the isomers under the used HPLC conditions has not yet been achieved. In Figure 6, an example of a comparison between the total ions chromatographic profiles of the HPLC-MS analysis in the *m*/*z* range between 250 and 750 of the ELF94 and ELF96 isomers is shown.

The peak parameters (i.e., retention times, peaks width, efficiency, etc.) for each analyte were calculated and reported in the Appendix A. The proposed HPLC method achieved poor efficiency (N ≈ 5000), mainly due to the short column used, but this setup allows to take advantage of a rapid elution gradient and a brief time to rinse the column and restore the initial condition; as a result, a faster chromatographic run is obtained.

### 2.4. LEDA Reliability

The reliability of LEDA elaboration was performed through the estimation of the accuracy and precision of the quantitative results obtained by processing the MS/MS data from the analysis of the mixture solutions of the isomers, prepared as described in Section 4.3. In fact, the algorithm “decodes” the acquired MS/MS signals, distinguishing the possibly present components and assigning their correct abundance. In this case, the integrated area of the chromatographic peak, corresponding to the elution of both isomers, was elaborated, the MS/MS signal was assigned, and the relative abundance of each component was calculated (ratio purity). As reported above, the IT acquires the MS/MS spectra in the product ions scan, then all the Pis above the 5% of abundance are involved in the LEDA matrix compilation (LEDA_All_). Hence, the used LEDA matrix was assembled by six Equation (4) (see Section 4.8), each related to a specific Pi selected to recognize the isomers (Appendix A. The performance of accuracy and precision of the LEDA algorithm was performed by analyzing five standard mixtures with different concentrations of isomers and by evaluating the related validation plots [22]. The validation plot was obtained by the graphical distribution of the results estimated from the LEDA algorithm with respect to the expected concentration values; then, a linear regression analysis was applied to find the best fitting between the data points. The calculated values of the linear function (e.g., slope, y-intercept, etc.) depict the accuracy and precision characteristics of the LEDA approach with a single value representative of all the concentration levels tested. The validation plots for both isomers are reported in Appendix A, while the obtained results are reported in Table 2.

All the slope values shown in Table 2 are close to 1, which represents a recognition accuracy near 100%, confirming the reliability of all employed LEDA matrices to distinguish the correct amount of each isomer present in standard mixtures. In addition, the precision features (estimated by SE-Lin values) were interesting; indeed, the purity ratio values ranged between 0.01 and 0.02, corresponding to an uncertainty of 1% and 2% of the isomer present in the mixture, respectively. Then, together with the y-intercept values near the origin, the assembled LEDA algorithm achieved a correct assignment of the MS/MS signal up to 2–4% of relative abundance of the isomer in the mixture (considering the 95% confidence interval such as ±2 SE-Lin). These results confirm that the LEDA tool is suitable for recognizing the studied isomers by elaborating MS/MS data from the IT mass analyzer system as well. Finally, to verify the proposed approach, the described HPLC-MS/MS method was applied in plasma stability studies of the selected isomers to evaluate the effectiveness of the algorithm also in a complex sample.

### 2.5. Chemical Stability Test

In a previous study, the quali-quantitative ability of the LEDA algorithm to support the evaluation of the hydrolytic activity of the human plasma enzymatic system when two potential substrates were simultaneously present was checked [22]. This investigation was planned to extend this approach at a different MS/MS system by passing from a “space” mass analyzer (such as QqQ) to a “time” mass spectrometer like the IT. The investigation involved the analysis of a series of human plasma samples spiked with an equimolar mixture of ELF94-ELF96 isomers. To monitor this process correctly, it must be possible to distinguish each isomer present in the sample, quantify it with the proper calibration curve, and plot its degradation profile. The plasma samples were prepared as described in Section 4.6 and analyzed with the proposed HPLC-MS/MS method. The acquired MS/MS data were processed with the LEDA tool that distinguished the relative concentration of the isomers present in each sample. Then, the isomers’ abundances were assigned by distributing them in the signal of precursor ion, the corresponding peak area ratio (PAR) value was calculated by dividing each isomer-assigned-peak area with the IS abundance, and quantitative determination of the analytes in the sample was performed by using the respective precursor ion calibration curve. In Figure 7, an example of degradation plots of the mixture of ELF94 and ELF96 isomers was shown.

The characteristic kinetic hydrolysis parameters (*t*_1/2_ and *k*) calculated from the analysis of the plasma samples spiked with the isomers mixture are reported in Table 3.

The obtained results confirmed the hydrolysis of the substrate isomer (ELF94) by plasma enzymes, while the concentration of the stable isomer (ELF96) remained constant in all samples tested. Also, the *t*_1/2_ and *k* values were comparable between the analyzers (QqQ and IT), despite the different MS/MS conditions used, demonstrating the effectiveness of the LEDA elaboration.

However, to further check the LEDA performances, the spiked plasma samples are processed with a ‘scan-by-scan’ approach, allowing the graphical separation of the isomers present in the chromatographic profiles. In this case, the computation of the LEDA matrix was repeated on each MS/MS data point acquired; consequently, a relative abundance of the Ri signal for each isomer was assigned. By considering a generic chromatographic profile, the performance of the algorithm was appreciated in the deconvolution of the unresolved peak of the Ri signal that delivered its abundance between the isomers (LEDA-reconstructed chromatographic profiles). In Figure 8, an example of LEDA-reconstructed chromatographic profiles of the human plasma sample spiked with the mixture of the ELF94 and ELF96 isomers at the incubation time of 0 min was shown.

The reconstructed profiles (Figure 8) showed an equal abundance of the isomers in the Ri signal as of the mixture solution added, demonstrating that at time 0, plasma enzymes cannot hydrolyze the substrates. Increasing the incubation times, the reconstructed profiles of ELF94 reported a decrease in peak intensity, while the peak of ELF96 maintained the same abundance (Appendix A). Therefore, the LEDA ‘scan-by-scan’ processing approach confirmed the data obtained in the previous elaboration (peak area integration) and demonstrated its reliability by distinguishing the abundance of each isomer present in the unresolved Ri signal.

## 3. Discussion

In the introduction of the manuscript, we proposed the question: is it possible to transfer the MS/MS data post-processing LEDA approach to another type of tandem mass spectrometry system? Then, we reported our experience in optimizing, developing, and evaluating the ion trap MS/MS system coupled with LEDA data elaboration in the isomers distinction.

The IT performs the experiments of tandem mass spectrometry using the same site, helium as collision gas, and different time steps to modify the conditions applied to studied ions. These operative characteristics produce some issues to solve for the LEDA application and result in comparison to the previous studies. The helium collision gas employed in each MS/MS experiment reported exhibits a very small cross-section to ensure a quick energization of the precursor ion. Then, different ExT were tested, allowing, in combination with increasing ExA, the activation of the fragmentation channels of the selected ion and the formation of this MS/MS spectrum. The investigation of the right combination between ExT and ExA and the efficiency of the CID process was carried out with a series of ERMS experiments. A large amount of data required the creation of many graphs, the evaluation of the decay of precursor ions (SY curves), and the introduction of a new parameter (PiY) to depict the yield of Pis formation (PiY curves). The analysis of these parameters led to the selection of the best CID condition by using ExA 35 a.u. at 50 ms of ExT. The general scheme-fragmentation pathway of the analytes, proposed by using the QqQ analyzer, is also confirmed in the IT-MS/MS spectra, but with some important differences. In fact, the MS/MS spectra of the isomers show a new fragment (Pi_7_) at 384 *m*/*z* and the loss of significance of Pi_2_ at 195 *m*/*z*, which appears in abundance ≤5%. These differences and the availability of all Pis signals in a defined range radically changed the setup of the LEDA algorithm, leading to the arrangement of an overdetermined system of linear equations. In our case, all Pis in analytes MS/MS spectra with abundance ≥5% were considered, managing a LEDA matrix of six equations. However, a shared signal between the isomers, with the same behavior in the used MS/MS procedure, must be chosen as Ri. To meet the required features, the highest intensity of the precursor ion, at an ExA value without the occurrence of any fragmentation, was selected as Ri. The acquisition of Ri and Pis signals, which characterize the isomers recognition, involves the sequential recording of two MS/MS events at different ExA and scan ranges. The MS/MS event 1, at low ExA, allows the acquisition of an unfragmented precursor ion (Ri signal), while the MS/MS event 2, with higher collision energy applied, records the Pis signals. To convert the signal intensity into quantitative data, the IS signal must also be monitored, and a further MS/MS event must be introduced. Unfortunately, each MS/MS event is time-consuming, and to ensure the proper sample frequency that correctly defines the interest chromatographic peaks, at least two time segments should be established to separate the IS from the isomers’ unresolved peak. The purpose is achieved using a longer HPLC column, optimizing the elution conditions with a compromise between peaks distinction and run time reduction. The HPLC-MS/MS method, coupled with the LEDA “mathematical device”, introduced to process the IT-MS/MS data, was validated by analyzing a series of known mixture solutions of ELF94-ELF96 isomers. The accuracy and precision of the quantitative ability of the LEDA elaboration were checked by plotting the “validation plot”. The obtained results showed an accuracy near 100%, with precision values ranging between 1% and 2%. Thus, the proposed approach proved to be suitable for studying the selected isomer compounds without their chromatographic separation but by applying and developing the MS/MS features. Finally, we applied the developed HPLC-MS/MS method to the analysis of a series of human plasma samples spiked with an equimolar mixture of both isomers. A comparison with data obtained in a previous study using a QqQ mass system showed no significant differences, demonstrating that the IT-MS/MS process, with proper development and setup, is suitable to support the investigations of mixtures of co-eluting isomers by ensuring their recognition.

## 4. Materials and Methods

### 4.1. Chemicals

The isomers ELF94 and ELF96 used in this study were synthesized, as reported by Dei et al. [23]. Acetonitrile (Chromasolv), formic acid, ammonium formate (MS grade), verapamil hydrochloride, and ketoprofen (analytical standard) were purchased by Merck (Milan, Italy). Ketoprofen ethyl ester (KEE) was obtained by Fisher’s reaction from ketoprofen and ethanol. Ultrapure water or mQ water (resistivity 18 MΩ cm) was obtained from Millipore’s Simplicity system (Milan, Italy). The human plasma was collected from healthy volunteers, pooled, and kept at −80 °C until use. All subjects gave their informed consent for inclusion before they participated in the study. The study was conducted following the Declaration of Helsinki, and the protocol was approved by the local Ethics Committee (Comitato Etico Regionale per la Sperimentazione Clinica della Regione Toscana, Sezione AREA VASTA CENTRO) Institutional Review Board (CE: 11166_spe, 11/09/2018 and CE: 10443_oss, 14/02/2017).

### 4.2. Instruments

The HPLC-MS/MS analysis was carried out using a Thermo LCQ Deca XP plus ion trap (Waltham, MA, USA) equipped with a Surveyor liquid chromatography system and an electrospray ion source (ESI). Raw data were collected and processed by Excalibur version 1.4 software. A thermostatic oven G-Therm 015 was used to maintain the samples at 37 °C during the degradation test, while the centrifuge Eppendorf 5415D was employed to centrifuge the plasma samples

### 4.3. Standard and Calibration Solutions

Stock solutions of analytes and verapamil hydrochloride (internal standard or IS) were prepared in acetonitrile at 1.0 mg mL^−1^ and stored at 4 °C. Working solutions of each analyte were freshly prepared by diluting stock solutions up to a concentration of 1.0 µg mL^−1^ and 0.1 µg mL^−1^ (working solutions 1 and 2, respectively) in a mixture of mQ water:acetonitrile 50:50 (*v*/*v*). The IS working solution was prepared in acetonitrile at 33 ng mL^−1^ (IS solution). The quantitative data of each analyte were obtained by building a five-level calibration curve prepared by adding proper volumes of working solution (1 or 2) of each analyte to 300 μL of IS solution. The obtained solutions were dried under a gentle nitrogen stream and dissolved in 1.0 mL of 10 mM of formic acid in mQ water:acetonitrile 80:20 (*v*/*v*) solution. The final concentrations of calibration levels of each analyte were: 5.0, 10.0, 25.0, 50.0, and 100.0 ng mL^−1^. In order to evaluate the accuracy and precision of the LEDA approach, five mixture solutions of a couple of the isomers in different proportions were prepared. Mixture 1 had 90% of A isomer and 10% of B isomer; in mixture 2, the proportion was 75% and 25%; mixture 3, 50% and 50%; mixture 4, 25% and 75%; and mixture 5, 10% and 90%. The sum of isomer concentrations in the five mixture solutions was 100 ng L^−1^. Each solution was used to prepare the precision and accuracy set samples as follows: 100 μL of mixture solution and 300 μL of IS solution in an autosampler vial were added. The obtained solutions were dried under a gentle nitrogen stream and dissolved in 1.0 mL of 10 mM of formic acid in mQ water:acetonitrile 80:20 (*v*/*v*) solution. The calibration levels and mixture solutions were analyzed six times by the HPLC-MS/MS proposed method. Finally, the spiked solution of the isomers mixture, used in the sample preparation of the chemical stability study, was separately prepared by diluting the respective stock solutions in mQ water:acetonitrile 80:20 (*v*/*v*) solution, to obtain the final concentration of 10 μM of each analyte.

### 4.4. MS and ERMS Experiments

The ESI source operates in a positive ion mode using the following setting: 5 kV source voltage, 35 arbitrary units (a.u.) sheath gas, 5 a.u. auxiliary gas, the capillary voltage was 40 V while its temperature was set at 280 °C and 20 V of the tube lens. The MS analyses were acquired in an ion scan by monitoring the *m*/*z* range from 250 to 750 with 50 ms of maximum ion trap fill time. The ERMS experiments were performed to study the fragmentation of [M+H]^+^ species of each analyte and build their breakdown curves [17,18,19,20,21,22]. The ERMS experiments were carried out by a series of product ion scan (MS/MS) analyses, increasing the ExA (named Normalized Collision Energy in the control panel of the instrument) stepwise in the range of 0–50 a.u. with different ExT (10, 25, 50, and 100 ms). Each acquired MS/MS spectrum was in the *m*/*z* range from 165 to 650 (q value 0.25) using helium as collision gas. The ERMS experiments were performed by introducing working solution 1 of each analyte (Section 4.3) via a syringe pump at 10 µL min^−1^; the protonated molecule was isolated, and the abundance of product ions was monitored. The obtained data were used to build the graphs that describe the energetic dimension of the CID process (SY curves of precursor ion, the Pis formation, and the Pis yield). The SY curve describes the energetics of degradation of precursor ions according to the growth of the ExA. Each SY value at a defined ExA has been calculated as follows:(1)$$SY(\%)=\frac{Precursor\;ion\;Abundance}{Precursor\;ion\;Abundance+\sum_{1}^{n}Pis\;Abundances}\times 100$$ Likewise, the Pis formation (PiF) curve describes the energetics of the formation of all Pis with the increment of applied ExA. Each PiF value at a defined ExA has been calculated as follows:(2)$$PiF(\%)=\frac{\sum_{1}^{n}Pis\;Abundances}{Precursor\;ion\;Abundance+\sum_{1}^{n}Pis\;Abundances}\times 100$$ The Pis yield (PiY) has been determined to depict the efficiency of the CID process and estimated by the ratio between the sum of abundances of the Pis at defined ExA and the average intensity of the precursor ion before its decay. Normally, the higher abundance of precursor ions was observed at low ExA values before the activation of fragmentation channels. The average of these abundances represents the quantity of the precursor ion signal (Precursor ion max) available for the CID process, while the sum of Pis abundances is the signal that remained. Then, the yield of the CID process should be described as follows:(3)$$PiY(\%)=\frac{\sum_{1}^{n}Pis\;Abundances}{Precursor\;ion\;max}\times 100$$ Finally, the CID graphs were plotted to describe the dependence of fragmentation/formation processes (breakdown curves) on the energy provided to the precursor ion. The breakdown curves were plotted by reporting the ratio between the intensity values (averaging 15–20 scans) of each ion signal in the MS/MS spectra versus the precursor ion max at the ExA applied. In this way, the ratio of the abundances represents the yield of formation of each Pi with respect to the precursor ion to the applied ExA.

### 4.5. HPLC-MS/MS Methods

The chromatographic parameters employed to analyze the samples were optimized to minimize the run time and to separate in two different time segments the IS peak from the analytes’ unresolved peaks. The column used was a Pursuit XRs C18 30 mm length, 2 mm internal diameter, and 3 μm particle size Agilent (Santa Clara, CA, USA), at a constant flow of 0.25 mL min^−1^, employing a binary mobile phases elution gradient. The used solvents were 10 mM formic acid and 5 mM ammonium formate in mQ water:acetonitrile 90:10 (solvent A) and 10 mM formic acid and 5 mM ammonium formate in acetonitrile:mQ water 90:10 (solvent B). The program of elution gradient was set up as follows: initiation at 90% solvent A, which was then decreased to 10% in 4.0 min, kept for 3.0 min, returned to initial conditions in 0.1 min, and maintained for 2.9 min for a total run time of 10.0 min. The column temperature was maintained at 40 °C, and the injection volume was 5 µL. The elaboration of data from ERMS experiments allowed the definition of the MS/MS conditions used to compile the final HPLC-MS/MS method. These conditions can be divided into common parameters, such as isolation width 3 *m*/*z*, 0.25 as q value, and ActT of 50 ms, or MS/MS event specifics, listed and reported in Table 4.

The first MS/MS event, applied in time segment 1 (between 0.00 and 3.27 min), acquires the IS signal, while the other two Pis scan events, alternately applied in the time segment 2 (between 3.28 and 10.00 min.), record the Ri and Pis signals of the isomers, respectively. The spectra of each analyte extracted from the MS/MS events, included in time segment 2 of the proposed HPLC-MS/MS method, are reported in Appendix A.

### 4.6. Sample Preparation

Each sample was prepared by the addition of 10 µL of the spiked solution of isomers mixture to 100 µL of human plasma matrix in a 1.5 mL microcentrifuge tube. The obtained solution corresponds to 1 μM of each analyte that undergoes degradation study. Each set of samples was prepared in triplicate and incubated for four different times: 0, 30, 60, and 120 min at 37 °C. Therefore, each stability panel of the studied analyte was represented by a batch of 12 samples (four incubation times three replicates). After the incubation, 300 µL of IS solution was added to the sample and then centrifuged (room temperature for 5 min at 800 g). Then, the supernatant was transferred in autosampler vials, dried under a gentle stream of nitrogen, and dissolved in 1.0 mL of 10 mM of formic acid in mQ water:acetonitrile 80:20 solution [28,29]. Following the proposed procedure, the human plasma matrix was diluted with a ratio 1:10 (i.e., 0.1 mL of human plasma at 1 mL final sample solution). Each sample batch also included the blank samples of the human plasma matrix, prepared as described above, but by only adding the IS solution. Thus, the analysis of the blank samples can check any interference in the analyte MS/MS signals due to the matrix components. The obtained solutions were transferred in autosampler vials, dried under a gentle nitrogen stream, and dissolved in 1 mL of 10 mM of formic acid solution in mQ water:acetonitrile 80:20. Following the procedure described above, the expected concentration of the samples was 60 ng mL^−1^, values placed in the middle of calibration curves. The final solutions of blank and all sets of samples, prepared as described above, were analyzed by the proposed HPLC-MS/MS method.

### 4.7. Calibration Curves of HPLC-MS/MS Methods

The proposed IT-MS/MS method is assembled with three Pis scan events, each one characterized by a different *m*/*z* range acquisition and specific energetic conditions. Time segments, MS/MS parameters, and the selected ionic signals for monitoring the IS and each isomer for quantitative determination are reported in Appendix A. The calibration curves of analytes were obtained by plotting the Ri and Pi_1_ peak area ratios (PARs), calculated between the analyte and IS quantitation ion signals versus the nominal concentration of each calibration solution. A linear regression analysis was applied to obtain the best-fitting function between the calibration points (Appendix A). Taking into account that both calibration and sample solutions were prepared with the same procedure, the calibration curve parameters, the LOD, and LOQ values for each analyte were referred to as the concentration of sample solution analyzed by the HPLC-MS/MS method. In order to obtain reliable LOD and LOQ values, the standard error (SE) of the response and slope approach was employed [30]. The estimated SEs of responses of each analyte were obtained by the SE of y-intercepts (Y-SE) of regression lines elaboration of data obtained from the HPLC-MS/MS analysis of calibration solutions [31]. The results of calibration curves obtained for MS/MS quantitation signals (Ri and Pi_1_), defined as linear regressions parameters (slope and y-intercept), the determination coefficient (R^2^), and the estimated LOD and LOQ values for each analyte are reported in Appendix A. The comparison among the calibration curves demonstrates how the slope values of the Ri signals are similar for both isomers and have higher respect for those obtained with the monitoring of Pi_1_. These differences can be related to the characteristics of fragmentation of each isomer; then, to maintain the same sensitivity and quantitative relevance, the determination of the studied analytes was performed by using the peak abundances assigned by LEDA elaboration on the Ri signal acquired from the sample [22].

### 4.8. LEDA Algorithm

The algorithm is based on the rationale that the MS/MS signal might be represented as the sum of the contribution of each isomer present in the sample. Therefore, it is reasonable to describe each Pi signal as the sum of the abundances of this ion coming from all the isomers eventually present in the sample. To manage reliable data and enhance the compound-dependent Pi yield differences, the relative abundances of selected Pis with respect to a reference ion (Ri) were calculated. The Ri signal must be chosen among the MS/MS signals with the same abundances at defined MS/MS conditions between the studied isomers. Unfortunately, none of the Pis detected in MS/MS spectra of analytes have the required features. In previous studies, the precursor ion signal was selected as Ri at its highest intensity in the breakdown curve without any fragmentation process occurring. By this approach, the ratio between the abundance of each Pi acquired, and the abundance of Ri represents the yield of the Pi formation at the selected collision voltage. In an IT application, using the signal of the precursor ion as Ri requires the insertion of a MS/MS event at low collision energy to enable the acquisition of an unfragmented precursor ion (MS/MS event 1). Naturally, in the same time segment, a second MS/MS event must be acquired, at relatively high collision energy, to obtain the related tandem mass spectra of the fragmented ions (MS/MS event 2). Therefore, knowing the characteristic abundance ratios of a pure isomer (Table 4, Section 4.8), a deconvolution of acquired MS/MS signals is possible based on a series of linear regression equations as follows:(4)$${\left(\frac{Pi}{Ri}\right)}_{m}=\sum_{x=1}^{n}{\left(\frac{Pi}{Ri}\right)}_{x}\cdot {\left[\%\right]}_{x}$$

(Pi/Ri)_m_ is the abundance ratio between the product ion (Pi) and reference ion (Ri) measured (m) in the sample;(Pi/Ri)_x_ is the characteristic abundance ratio between the Pi and Ri of pure isomers;[%]_x_ is the concentration (%) of the isomer in the sample.

Theoretically, considering a simple binary mixture of isomers (A-B), a single Equation (4) related to only a product ion ratio (Pi/Ri) could be sufficient. Indeed, by assuming that only the pair of isomers constitutes the MS/MS signal, the concentration of B is calculated as B% = (1 − A)%. However, in this case, the possible contribution of signals from unknown isomers (or any co-eluting compound having the same product and reference ions) is neglected. Therefore, for the MS/MS signal distinction of mixtures of n isomers, it is preferable to use a matrix with n linear Equation (4) (LEDA algorithm). Naturally, to increase the specificity and reliability of isomers speciation, an overdetermined system of linear equations could be assembled; in this case, the LEDA matrix was composed of a number >n of linear equations (see Equation (4)). This is the case of IT that operates in MS/MS acquiring the Pis in a defined range of *m*/*z* (product ions scan); then, all Pis signals included in the range are recorded and can be used to set up the LEDA matrix. In this way, considering only the Pi signals ≥5%, an overdetermined system of linear equations (six equations) is assembled (Appendix A). The deconvolution was performed by applying the algorithm either to the area abundances obtained from the integrated peak intensities of each Pi signal or to individual MS/MS data points of the chromatographic sample profile. In the first case, the LEDA provides the relative amounts (%) of each known component present in the sample. In the second approach, each MS/MS signal is deconvoluted ‘scan-by-scan’ and assigned to the present isomers, allowing a graphical separation of the processed chromatographic profiles. All calculations for the deconvolution of MS/MS data are processed using an Excel™ macro. The characteristic abundance ratios were calculated by data obtained from the highest level of the calibration curve (100 ng mL^−1^) of each pure isomer by the HPLC-MS/MS methods described above. The ratios between Pi and Ri selected in the acquired signals were calculated, and the resulting values were reported in Table 5.

The LEDA tool performances were evaluated by processing the MS/MS data obtained from the analysis of the standard mixture samples prepared in Section 4.3, and the results were checked by comparison with the expected values.

### 4.9. Chemical Stability Test

In general, the presence of an ester group in the structure of a new drug candidate compound requires a plasma stability study to verify its possible hydrolysis by esterase enzymes. The plasma samples were prepared by adding a known amount of compound (usually between 1 μM and 5 μM), analyzed by a proper method, and finally, their stability profiles were plotted. The stability profile of each compound was obtained by monitoring the variation of the analyte concentration at different incubation times in plasma samples (0, 30, 60, 120 min). Generally, when the substrate concentration was smaller than the Michaelis–Menten constant (K_M_), the enzymatic degradation rate is described as first-order kinetics. Therefore, by converting the quantitative results as natural logarithms, the natural logarithm of concentrations can be plotted versus incubation times and the slope of the linear function obtained will represent the degradation rate constant (*k*). Then, the half-life (*t*_1/2_) of each tested compound can be easily calculated as follows:(5)t12=ln (0.5 μM)k

For the compounds that showed a value of *k* rate < 0.006 (ln(μM)/min.), a high *t*_1/2_ value will be determined. However, considering the measurement errors and that the highest incubation time involved in the study was 120 min, the *t*_1/2_ parameter of compounds with a low *k* rate has been calculated up to the limit value of 120 min and beyond has been indicated as >120 min. The hydrolytic activity of the employed pool of human plasma was checked by adding KEE (reference compound), and after application of the same procedure followed for the studied compounds, its degradation plot (Appendix A) showed a decrease comparable to that found in the literature (*t*_1/2_ ≤ 120 min), confirming the activity of plasma enzymes [26].

## Figures and Tables

**Figure 1 ijms-24-11155-f001:**
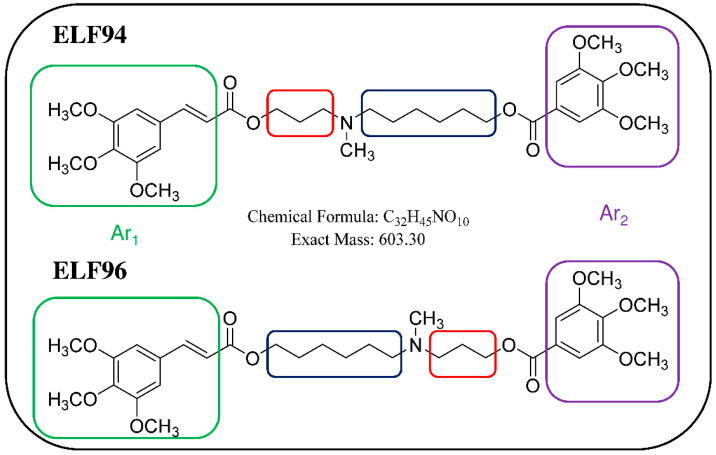
Chemical structures of positional isomers selected for this study. Red and blue squares highlight the structural differences between the compounds, while the green and purple squares enclose the aromatic moieties (Ar1 and Ar2, respectively).

**Figure 2 ijms-24-11155-f002:**
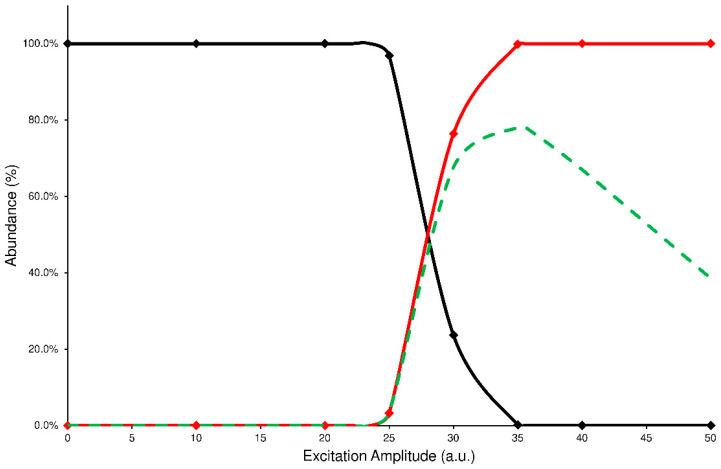
Precursor SY curve (black line), PiF (red line), and PiY (green dashed line) of ELF94 isomer at ExT 50 ms.

**Figure 3 ijms-24-11155-f003:**
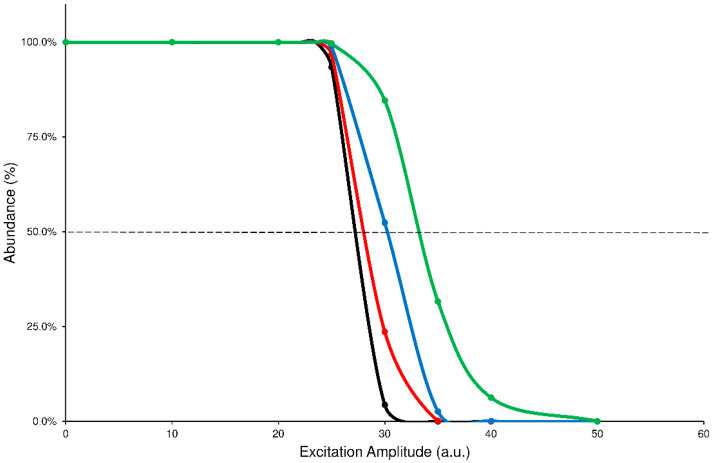
Comparison between precursor SY curves of ELF94 by using ExT values of 10 ms (green line), 25 ms (blue line), 50 ms (red line), and 100 ms (black line).

**Figure 4 ijms-24-11155-f004:**
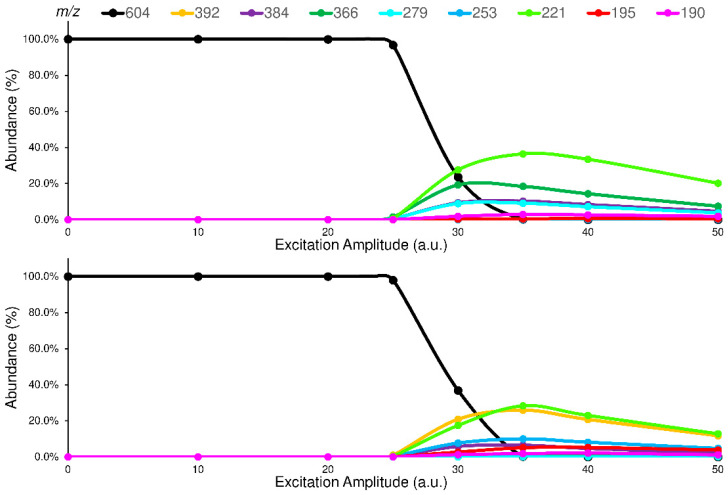
Comparison between the breakdown curves of ELF94 (**top**) and ELF96 (**bottom**) isomers.

**Figure 5 ijms-24-11155-f005:**
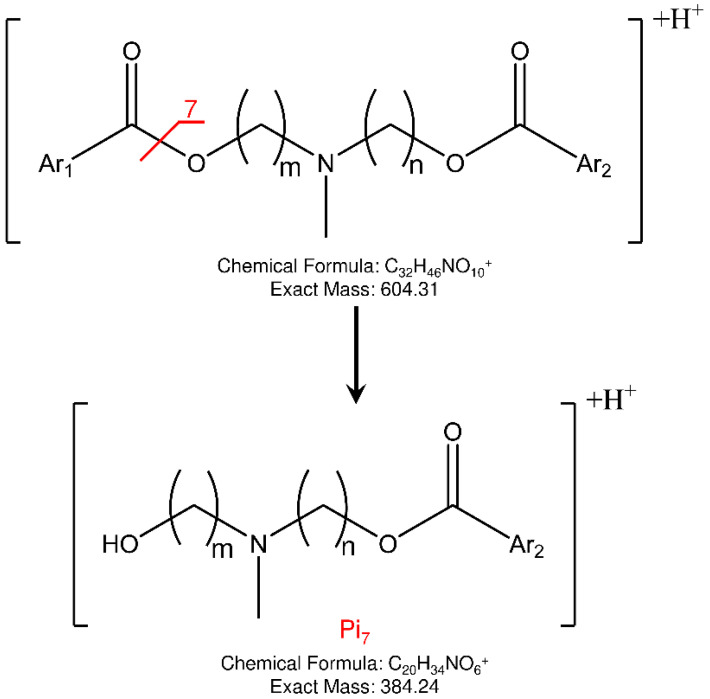
Proposed fragmentation and structure of new product ion (Pi_7_) observed in IT-MS/MS spectra of ELF94 and ELF96 isomers.

**Figure 6 ijms-24-11155-f006:**
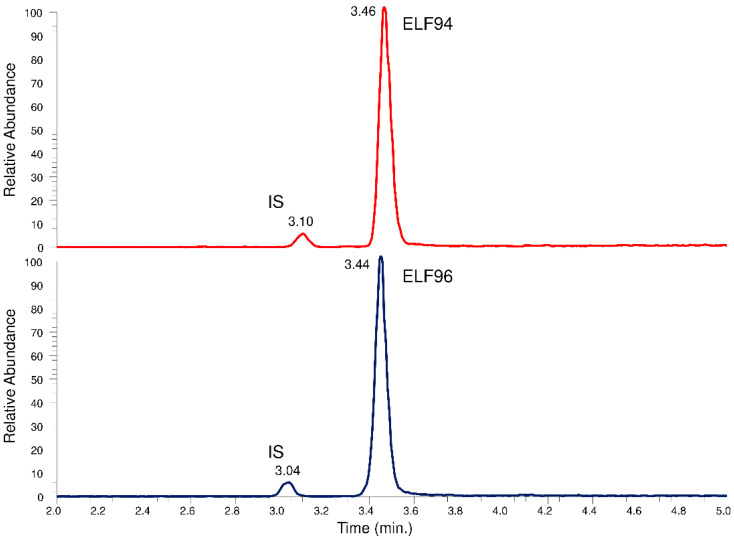
Total ions chromatographic profiles of the HPLC-MS analysis of the ELF94 and ELF96 isomers.

**Figure 7 ijms-24-11155-f007:**
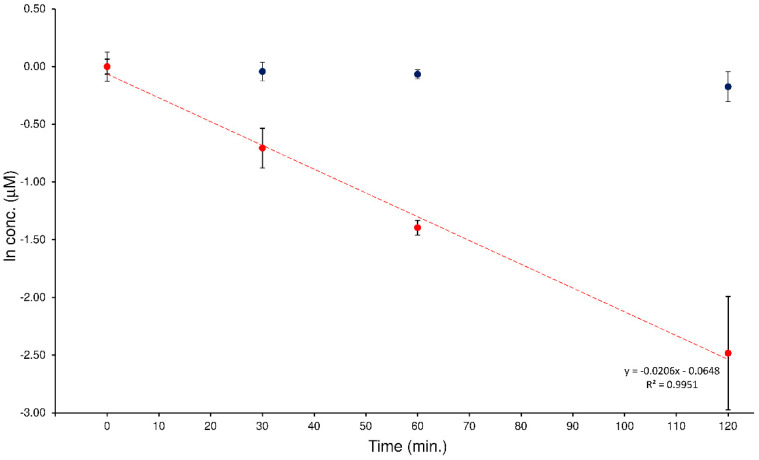
Degradation plots obtained by LEDA elaboration of the human plasma samples spiked with a mixture of the ELF94 (red circles) and ELF96 (blue circles) isomers.

**Figure 8 ijms-24-11155-f008:**
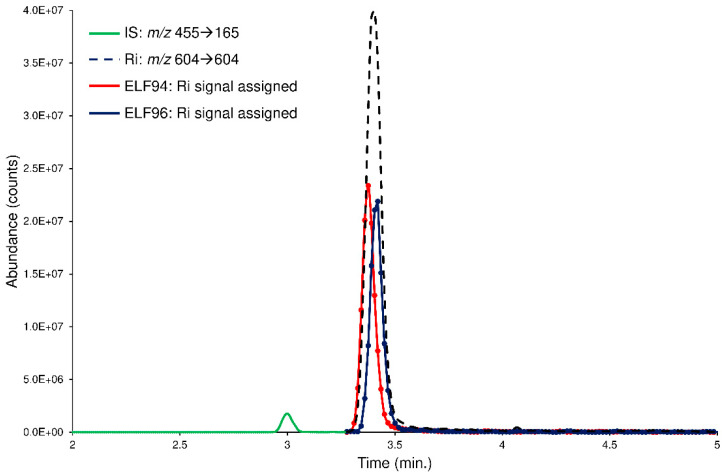
The LEDA-reconstructed chromatographic profiles of the human plasma sample spiked with the mixture of the ELF94 (red line) and ELF96 (blue line) isomers at the incubation time of 0 min. The IS (green line) and Ri (black dotted line) signals were also reported.

**Table 1 ijms-24-11155-t001:** The efficiency of the CID process (PiYmax) and related ExA applied (ExA_max_) obtained from ERMS experiments carried out on ELF94 and ELF96 isomers.

	ELF94	ELF94	ELF96	ELF96
ExT	PiY_max_	ExA_max_	PiY_max_	ExA_max_
(ms)	(%)	(a.u.)	(%)	(a.u.)
10	57.5	40	62.9	40
25	78.1	35	75.1	35
50	78.9	35	77.7	35
100	80.0	35	75.4	35

**Table 2 ijms-24-11155-t002:** Validation plot parameters (slope, y-intercept, determination coefficient or R^2,^ and standard error of linear function or SE-Lin) obtained by the elaboration of MS/MS data of standard mixtures of ELF94 and ELF96 isomers by the LEDA_All_ matrix described above.

	Slope	y-Intercept	R^2^	SE-Lin
ELF94 LEDA_All_	0.98	0.01	0.998	0.02
ELF96 LEDA_All_	0.99	−0.03	0.999	0.01

**Table 3 ijms-24-11155-t003:** Comparison of the characteristic degradation parameters, half time (*t*_1/2_), and degradation rate (*k*), with the estimated standard deviations (SD), calculated by LEDA processing the data from the spiked human plasma samples by using QqQ and IT systems.

	QqQ (*)	QqQ (*)	IT	IT
	*t*_1/2_ ± 2SD	*k* ± 2SD	*t*_1/2_ ± 2SD	*k* ± 2SD
	(min.)	(ln(μM)/min.)	(min.)	(ln(μM)/min.)
ELF94	34 ± 14	−0.023 ± 0.005	31 ± 13	−0.021 ± 0.007
ELF96	>120	<0.006	>120	<0.006

(*) Data reported in ref [22].

**Table 4 ijms-24-11155-t004:** Time segments and MS/MS parameters used for the acquisition of IS and the studied isomers.

Compound	Time Segment	Time Segment (min.)	Precursor Ion (*m*/*z*)	MS/MS Event	Pis scan Range (*m*/*z*)	ExA (a.u.)
IS	1	0.00–3.27	455	IS	135–350	35
ELF94 ELF96	2	3.28–10.00	604	Ri	580–620	20
				Pis	165–450	35

**Table 5 ijms-24-11155-t005:** Characteristic ion abundance ratios (Pi/Ri) ± standard deviation (SD) calculated by MS/MS data obtained from the analysis of the 100 ng mL^−1^ solution of each pure isomer by HPLC-MS/MS methods described in Section 4.4.

Pair of Isomers	Ratio Pi/Ri (*m*/*z*)	ELF94 Ratio Value ± SD	ELF96 Ratio Value ± SD
ELF94-ELF96	392/604	0.01 ± 0.01	0.27 ± 0.01
	384/604	0.09 ± 0.01	0.07 ± 0.01
	366/604	0.19 ± 0.01	0.01 ± 0.01
	279/604	0.10 ± 0.01	0.01 ± 0.01
	253/604	0.01 ± 0.01	0.11 ± 0.01
	221/604	0.38 ± 0.01	0.28 ± 0.01

## Data Availability

Not applicable.

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
