# Peer review of "Isomers Recognition in HPLC-MS/MS Analysis of Human Plasma Samples by Using an Ion Trap Supported by a Linear Equations-Based Algorithm"

_ijms, 2023, doi:10.3390/ijms241311155_

Round 1

Reviewer 1 Report

General comments:

This paper describes a nice application of the analysis of co-eluted isomers by energy-resolved mass spectrometry. To this aim, the authors apply the LEDA algorithm. This algorithm has been applied by the authors before for the analysis of several isomeric compounds. The novelty, in this case, is that they show the feasibility of applying LEDA in the context of an ion trap instrument. All the applications described before were performed in QqQ instruments (widely used in chromatography) LEDA allows the correct quantification of both isomers. However, both isomers show specific transitions. ELF96 shows two specific fragments: 392 m/z and 253 m/z. And ELF94 shows also two specific transitions: 366 m/z and 279 m/z. This means that MRM could be applied without the need of using LEDA and deconvolution of the chromatograms. Extract Ion chromatograms could be obtained for both co-eluted isomers. Even, one of the fragments (392m/z) could be used as target ion and 253 m/z as qualitative ion (less intensity). The same for ELF94. 366 m/z could be the target ion and 279 m/z would be the qualitative ion. I think that the authors should compare the results obtained by LEDA with the ones of classical MRM (widely used in chromatography). What is the added value of applying LEDA compared to classical MRM? I think that LEDA is necessary when there are not specific fragments. But in this case, there are specific fragments for both isomers. A comparison of the performance parameters obtained in MRM and LEDA for the analysis of both isomers should be shown in the manuscript. Moreover, it would be interesting to show the performances of LEDA in the case where there are no specific fragments. In this case, obtaining good quantitative results is more challenging than in the case showed. (from 6 fragments, 4 are specific). It would be interesting also if the authors show what is the added value of using an Ion Trap instead of a QqQ. It is stated the disadvantage of Ion Trap: more time to measure than in QqQ and the necessity of separating the internal standard from the isomers in order to do two acquisition windows. However, what is the advantage of Ion Trap compared to QqQ? This maybe could be more clearly explained in the paper. To justify the use of ion trap instead of QqQ.

Specific comments:

There are some mistakes to correct in English. Without being exhaustive, some mistakes to correct are:

• Page 5, line 161. “are report” should be corrected by “are reported”

• Page 3, line 112. “A typical graphs used to describe…” should be maybe “a typical graph”

• Page 8, line 220. “the MS/MS data from the analyzes of the mixture”. It should be corrected “the MS/MS data from the analysis of the mixture”

• Page 13. Line 423 “The SY curve describe” should be “describes”

• Supporting Information. Legend ST1 “for each analyte obtained apply the proposed HPLC-MS/MS approach.” Correct maybe by “obtained with the proposed…”

Validation of LEDA. Table 2 (page 8). And support info Figure SF19. For the isomer ELF96, the intercept is -0.03. Is there a negative constant bias in the quantification of this isomer of about 3%? Usually, significance tests of the slope and the intercept are performed in order to evaluate the absence of significant constant and proportional bias.

The LOD is calculated from the calibration curve (standard error divided by the slope). However, the human serum samples are diluted ten times for the chromatographic analysis. This means that the LOD and LOQ should be multiplied by 10, if the method is applied in serum samples.

Internal standard calibration curve. Some explanations are given in section 4.7. and also in page 9: “Then, the isomers abundances were assigned by dis- 267 tributing them in the signal of precursor ion (or Ri signal), the corresponding PAR value 268 calculated, by dividing with the IS abundance, and quantitative determination of the analytes in the sample performed by using the respective precursor ion calibration curve” And the results from the calibration curve are shown in supporting info (table ST2).

I think the authors should explain more clearly the calculations. From the paragraph in page 9, it is possible to guess that the intensity plotted for each analyte corresponds probably to the precursor ion measured at excitation voltage of 20 V (before fragmentation of the precursor ion). But, what is the intensity measured for the internal standard? Table 4 (page 14) shows only that IS (455 m/z) was measured at 35 V and only the fragments were measured (between 135 and 350 m/z). The intensity of the IS corresponds to the total intensity of the fragments of IS? Or is there an error in Table 4, and also the IS was measured at 20 V to have the intensity of the precursor ion of IS?

Moreover in supporting info, table ST2 shows two calibration curves for both isomers. One for Pi and the other one for Ri. What does it mean Pi, in this case? The name of the column is MRM transition. So, one could think that Pi corresponds only to one transition? Is this the case? If yes, which transition? And if not, does it correspond to the sum of all the 6 selected product ions? I think that this part should be better explained. This is quite important, because the importance of this paper is focused on the quantification of the isomeric compounds. Also, a figure with the calibration curves in the supporting info could be included.

Author Response

Dear Editor,

I received the Reviewers’ comments to the manuscript "Isomers recognition in HPLC-MS/MS analysis of human plasma samples by using ion trap supported by LEDA data elaboration tool" and I wish to thank for their useful suggestions.

Enclosed please find the revised version of the manuscript amended according to their suggestions.

A point-wise list of response to the comments and the corrections (red text), now enclosed in the paper, is attached to this message.

I hope that in present form the paper could be considered worthy of publication.

Thank you for the opportunity you gave us to resubmit the paper.

Waiting for again kind answer, I remain

Sincerely yours

Gianluca Bartolucci, (corresponding Author)

Response to Reviewer 1

General comments:

This paper describes a nice application of the analysis of co-eluted isomers by energy-resolved mass spectrometry. To this aim, the authors apply the LEDA algorithm. This algorithm has been applied by the authors before for the analysis of several isomeric compounds. The novelty, in this case, is that they show the feasibility of applying LEDA in the context of an ion trap instrument. All the applications described before were performed in QqQ instruments (widely used in chromatography) LEDA allows the correct quantification of both isomers.

However, both isomers show specific transitions. ELF96 shows two specific fragments: 392 m/z and 253 m/z. And ELF94 shows also two specific transitions: 366 m/z and 279 m/z.

The authors would like to elaborate the reviewer's sentence "However, both isomers show specific transitions". In present paper and in ref. 22 (QqQ study), the authors never report "specific transitions or specific product ions" but described the MS/MS spectra as formed by common product ions for both isomers with ".... different relative abundances". Therefore, each isomer, in the same MS/MS condition, produces the same pattern of product ions but with different branching ratios. Therefore, each isomer, in the same MS/MS condition, produces the same pattern of product ions but with different branching ratios. In our opinion, the reviewer's misunderstanding arises by the comparison between the MS/MS spectra of isomers (SF24-SF25) and/or the data of the characteristic ion abundance ratios reported in table 5. The "specific transitions" mentioned by the reviewer show a favored yield of formation (respect to the precursor ion) for an isomer over the other but does not exclude the mutual interference. How much can this interference affect the acquired signal? The case of the 392 m/z product ion that shows the yield of formation of 1% and 27 % (SD 1 %) for ELF94 and ELF96 respectively. But the isomers are simultaneously present in the sample and are co-eluted by the chromatographic column. Then, by taking a 90:10 mixture of isomers the 392 m/z signal is the sum of :

ELF94, 0.9 x 0.01 = 0.009;

ELF96, 0.1 x 0.27= 0.027;

Hence, the 25% (0.009/0.036) of the acquired 392 m/z signal are coming from the ELF94 isomer, that showing the lowest of yield of formation for this product ion.

This means that MRM could be applied without the need of using LEDA and deconvolution of the chromatograms. Extract Ion chromatograms could be obtained for both co-eluted isomers. Even, one of the fragments (392m/z) could be used as target ion and 253 m/z as qualitative ion (less intensity). The same for ELF94. 366 m/z could be the target ion and 279 m/z would be the qualitative ion. I think that the authors should compare the results obtained by LEDA with the ones of classical MRM (widely used in chromatography).

Joining with the previous answer, the application of the classical MRM procedure must provide the separation of the isomers (in different sample or by chromatography) to avoid mutual signal interferences.

What is the added value of applying LEDA compared to classical MRM?

The use of a methodological approach that simplifies the liquid chromatography (HPLC) parameters, allowing the use of a short column and a fast elution gradient, leading to an increased productivity without losing determination specificity. With this approach, the chromatographic column was used only to avoid or limit the interference of the sample matrix towards the analyte ionization process (matrix effects). Then, the MS/MS properties were explored to solve the signal assignment.

I think that LEDA is necessary when there are not specific fragments. But in this case, there are specific fragments for both isomers. A comparison of the performance parameters obtained in MRM and LEDA for the analysis of both isomers should be shown in the manuscript. Moreover, it would be interesting to show the performances of LEDA in the case where there are no specific fragments. In this case, obtaining good quantitative results is more challenging than in the case showed. (from 6 fragments, 4 are specific).

The authors believe that this point has been explained in the previous sentences.

It would be interesting also if the authors show what is the added value of using an Ion Trap instead of a QqQ. It is stated the disadvantage of Ion Trap: more time to measure than in QqQ and the necessity of separating the internal standard from the isomers in order to do two acquisition windows. However, what is the advantage of Ion Trap compared to QqQ? This maybe could be more clearly explained in the paper. To justify the use of ion trap instead of QqQ.

The authors have proposed the question: "Is it possible to transfer the MS/MS data post-processing LEDA approach to another type of tandem mass spectrometry system?". Then, the purpose is not finding "....the added value of using an Ion Trap instead of a QqQ" but if the transfer is feasible. The opinion of the authors that the aim was clearly explained in the paper (section 2: results and section 3: discussion).

Specific comments:

There are some mistakes to correct in English. Without being exhaustive, some mistakes to correct are:

  • Page 5, line 161. “are report” should be corrected by “are reported”

The authors corrected by following the reviewer suggestion.

  • Page 3, line 112. “A typical graphs used to describe…” should be maybe “a typical graph”

The authors corrected by following the reviewer suggestion.

  • Page 8, line 220. “the MS/MS data from the analyzes of the mixture”. It should be corrected “the MS/MS data from the analysis of the mixture”

The authors corrected by following the reviewer suggestion.

  • Page 13. Line 423 “The SY curve describe” should be “describes”

The authors corrected by following the reviewer suggestion.

  • Supporting Information. Legend ST1 “for each analyte obtained apply the proposed HPLC-MS/MS approach.” Correct maybe by “obtained with the proposed…”

The authors corrected by following the reviewer suggestion.

Validation of LEDA. Table 2 (page 8). And support info Figure SF19. For the isomer ELF96, the intercept is -0.03. Is there a negative constant bias in the quantification of this isomer of about 3%? Usually, significance tests of the slope and the intercept are performed in order to evaluate the absence of significant constant and proportional bias.

The authors proposed the use of validation plot only to evaluate the quantitative performance of LEDA, then it is not used as calibration curve.

The LOD is calculated from the calibration curve (standard error divided by the slope). However, the human serum samples are diluted ten times for the chromatographic analysis. This means that the LOD and LOQ should be multiplied by 10, if the method is applied in serum samples.

The authors are conscious of reviewer's remark, but the proposed application does not assay the residues of studied analytes eventually present in unknown plasma samples. The samples were spiked with a known amount of analyte (1 mM), then the target of this study was the evaluation of the chemical stability of each analyte in experimental conditions (i.e. it maintains the initial concentration or decrease). Taking into the account of the aim of the study, the authors considered the parameters of quantitative determination (slope, intercept, LOD, etc..) of each analyte in the sample solution actually analyzed by the HPLC-MS/MS methods. Therefore, they believe that the reported values of LOD are better suited to represent the detection characteristics of the HPLC-MS/MS methods for this purpose.

In fact, to clarify this topic, the authors reported in the 4.7 section the sentence as follows: “Taking into the account that both calibration and sample solutions were prepared with the same procedure, the calibration curve parameters and the LOD value for each analyte were referred to the concentration of sample solution actually analyzed by the HPLC-MS/MS methods.”

Internal standard calibration curve. Some explanations are given in section 4.7. and also in page 9: “Then, the isomers abundances were assigned by distributing them in the signal of precursor ion (or Ri signal), the corresponding PAR value calculated, by dividing with the IS abundance, and quantitative determination of the analytes in the sample performed by using the respective precursor ion calibration curve” And the results from the calibration curve are shown in supporting info (table ST2).

The authors are rephrased the sentence in page 9 as follows: “Then, the isomers abundances were assigned by distributing them in the signal of precursor ion (or Ri signal), the corresponding peak area ratio (PAR) value calculated, by dividing each isomer assigned peak area with the IS abundance, and quantitative determination of the analytes in the sample performed by using the respective precursor ion calibration curve”.

I think the authors should explain more clearly the calculations. From the paragraph in page 9, it is possible to guess that the intensity plotted for each analyte corresponds probably to the precursor ion measured at excitation voltage of 20 V (before fragmentation of the precursor ion). But, what is the intensity measured for the internal standard? Table 4 (page 14) shows only that IS (455 m/z) was measured at 35 V and only the fragments were measured (between 135 and 350 m/z). The intensity of the IS corresponds to the total intensity of the fragments of IS? Or is there an error in Table 4, and also the IS was measured at 20 V to have the intensity of the precursor ion of IS? Moreover in supporting info, table ST2 shows two calibration curves for both isomers. One for Pi and the other one for Ri. What does it mean Pi, in this case?

The authors agree with the reviewer suggestion and added a new table in supplementary materials (ST2) and in section 4.7 the following sentences: “The proposed IT-MS/MS method is assembled with three Pis scan events, each one characterized by a different m/z range acquisition and specific energetic conditions. Time segments, MS/MS parameters and the selected of ionic signals for monitoring the IS and each isomer for quantitative determination are reported in supplementary materials (ST2)”. “The comparison among the calibration curves demonstrates how the slope values of the Ri signals are similar for both isomers and higher respect to those obtained with monitoring of Pi1. This differences can be related to the characteristics of fragmentation of each isomer then, to maintain the same sensivity and quantitative relevance, the determination of the studied analytes was performed by using the peak abundances assigned by LEDA elaboration on Ri signal acquired from the sample [22]”. The old ST2 in supplementary materials, now is ST3.

The name of the column is MRM transition.

The authors renamed the column in ST3 as “MS/MS signal”

So, one could think that Pi corresponds only to one transition? Is this the case? If yes, which transition? And if not, does it correspond to the sum of all the 6 selected product ions? I think that this part should be better explained. This is quite important, because the importance of this paper is focused on the quantification of the isomeric compounds.

The authors believe that this point has been explained in the previous sentences.

Also, a figure with the calibration curves in the supporting info could be included.

The authors, following the reviewer suggestion, added the figures SF26-SF29 of the calibration curves in supplementary materials.

Reviewer 2 Report

How to fix the number of components to perform the deconvolution , in real life such information is missing

how LEDA compares with other techniques likes multivariate curve resolution  (MCR-ALS) or Non -Negative matrix factorization 

the impact of baseline offset and signal to noise ratio

figure 7 make separate plot for both the isomers 

English is okay

Author Response

Dear Editor,

I received the Reviewers’ comments to the manuscript "Isomers recognition in HPLC-MS/MS analysis of human plasma samples by using ion trap supported by LEDA data elaboration tool" and I wish to thank for their useful suggestions.

Enclosed please find the revised version of the manuscript amended according to their suggestions.

A point-wise list of response to the comments and the corrections (red text), now enclosed in the paper, is attached to this message.

I hope that in present form the paper could be considered worthy of publication.

Thank you for the opportunity you gave us to resubmit the paper.

Waiting for again kind answer, I remain

Sincerely yours

Gianluca Bartolucci, (corresponding Author)

Response to Reviewer 2

Comments and Suggestions for Authors

How to fix the number of components to perform the deconvolution, in real life such information is missing

The aim of this work was to evaluate the stability of positional isomers when they’re simultaneously present in human plasma. So, in the beginning, we have to study them singularly by ERMS experiments and then we perform the stability tests. In an unknown sample we have to know the isomers composition present in the sample (i.e. separating them chromatographically and performing ERMS experiments to study each fragmentation pattern) and then, by using a faster program let the algorithm distinguish them.

how LEDA compares with other techniques likes multivariate curve resolution  (MCR-ALS) or Non -Negative matrix factorization

The authors developed LEDA algorithm to be specifically applied to the recognition of co-eluting isomers using Pi/Ri ratios (so using MS/MS experiments by employing multistage Mass Spectrometry). So far, we did not compare with the techniques mentioned by the reviewer, but we are aware they have a lot of similarities with LEDA (both using matrices to distinguish signal composed by two or more components).

the impact of baseline offset and signal to noise ratio

The authors have found that the evaluation of the baseline and s/n ratio were not necessary for the development of this work since the good performances obtained from the MS/MS method. In fact, we focused on the variation of the analytical signal by analyzing the calibration levels to evaluate the LOD and LOQ values (section 4.3 refs 27-28).

figure 7 make separate plot for both the isomers

In this case the authors decided to leave the degradation plots together because we studied them simultaneously in human plasma. In our opinion, the combined plot used for this study is more coherent in showing the data obtained, believing that are not misleading risks.

Reviewer 3 Report

The authors have developed an algorithm called LEDA that is designed to separate the MS signals of structural isomers that do not yield single, specific fragment ions. This allows them to quantify individual isomers from mixtures without the need for prior chromatographic separation. 

The authors have validated LEDA in previous publications using QqQ MS. Here, the authors apply the algorithm to ion trap MS.

The manuscript therefore describes the application of an existing technique to a different MS analyzer. Given the interest in isomer separations, this is a warranted goal. However, this raises the bar of what should be expected from the manuscript in terms of quality and understandability. As of yet, the manuscript is difficult to understand for readers unfamiliar with LEDA already. This is due to a combination of i) a need for English language editing; ii) many explanations required for understanding the results section (including the definition of abbreviations) are presented in the methods section; and iii) the figures not being furnished with legends (m/z, or otherwise).

General remarks

Figures: The figures are provided in excellent quality/resolution. They would benefit greatly from legends explaining what EXACTLY is being shown (e.g. which m/z is plotted). Also, the concepts behind LEDA and the difference in mass analyzers used in this study (IT) compared to previous studies (QqQ) could be presented visually. This is hinted at in the comment to Fig. 2.

Self-citations: more than 39% of all references are self-citations (11/28). This is understandable, since LEDA was developed by the authors themselves, but nevertheless may raise concerns. 

Specific comments

The title contains an uncommon abbreviation (LEDA).

34: MS is not necessarily "high-throughput." In many laboratories it is the bottleneck due to there being only one MS, while for many other instruments (e.g., HPLC-UV) multiple units are available.

52: "Specificity" has a specific meaning in analytical chemistry. What is the meaning of the phrase "determination specificity" used here?

60: Please specify what "large" in the phrase "relatively large collision gas (Argon)" refers to (diameter?).

91: The abbreviation "MSn" needs to be defined (sequential MS). Throughout the manuscript, many abbreviations are used prior to their definition. An additional section ("abbreviations") might be helpful to collectively define the many abbreviations used throughout the manuscript (in addition to their definition within the text).

111: The term "product ions" is used twice, but once abbreviated (Pis) and once spelled out.

Fig. 2: How could SY and PiF reach values greater than 100% as depicted by the fitted line? The denominator is the sum of two positive terms, one of them being the numerator.

Also, the concept of LEDA applied here to IT is currently difficult to understand for readers not familiar with its application to QqQ, i.e. readers unfamiliar with [22] or other manuscripts by the authors. Hence, the following would be helpful additions (subfigures) to Fig. 2:

A) A graphical visualization of the principle behind single-site fragmentation in ion trap MS contrasting "space" mass analyzer (QqQ) with "time" mass analyzer (IT);

B) A visualization of what SY, PiF, and PiY correspond to (in addition to including a verbal explanation near Fig. 2, since the mathematical explanations are currently provided in the much later section 4.4).

C) m/z or MS spectra of the observed SY, PiF, and PiY (e.g., SF12).

Fig. 5: Please add the m/z of the precursor and product ion to emphasize them being the same for ELF94 and ELF96.

193: Abbreviation "Ri" used but not defined until a later section. The same is true for other abbreviations such as PAR (line 268).

Figure 6: Please label which chromatogram shows which compounds, as well as which peak corresponds to IS and which to ELF94/ELF96, so the reader does not have to be refered to ST1.

213: What was the void time of the chromatographic system? This information is necessary to judge if the retention of IS and ELF94/ELF96 is sufficient to separate them from potential matrix components.

Table 2: Even though it here refers to a validation plot of "expected vs. obtained" values, R2 by itself has been deprecated as sole measure of a calibration model's fit (e.g., Félix Camirand Lemyre et al., Squaring Things Up with R2: What It Is and What It Can (and Cannot) Tell You, Journal of Analytical Toxicology, https://doi.org/10.1093/jat/bkab036). Please add the accuracy and precision values obtained for each calibrator.

Please also add the accuracy and precision values for at least one set of QC controls.

247: Given its detailed discussion (including the importance of the y-intercept near the origin), it may be helpful to show the validation plot in the manuscript and not in the supplementary information.

Fig. 7: Why is the curve fitted only for ELF94 but not for ELF96? A slope near 1 would show its stability. Also, the labels given in the caption are incorrect (red circles instead of pink squares).

Fig. 8: Where possible, please add the m/z legend for each trace to the figure to delineate single m/z traces from LEDA traces. This would be helpful also in previous chromatograms.

303: "did not work" might be re-phrased to not invoke the impression of the experiment having been unsuccessful. The fact that no discernible degradataion took place at time 0 was to be expected, and indeed is used here as a proof of concept.

Section 4.3.: Were the final concentrations given in the text post-reconstitution? What was the final concentration of the IS?

Section 4.7.: What were the concentrations of each calibrator pre-sample preparation?

Extensive editing of the English language would be benefical.

Author Response

Dear Editor,

I received the Reviewers’ comments to the manuscript "Isomers recognition in HPLC-MS/MS analysis of human plasma samples by using ion trap supported by LEDA data elaboration tool" and I wish to thank for their useful suggestions.

Enclosed please find the revised version of the manuscript amended according to their suggestions.

A point-wise list of response to the comments and the corrections (red text), now enclosed in the paper, is attached to this message.

I hope that in present form the paper could be considered worthy of publication.

Thank you for the opportunity you gave us to resubmit the paper.

Waiting for again kind answer, I remain

Sincerely yours

Gianluca Bartolucci, (corresponding Author)

Response to Reviewer 3

Comments and Suggestions for Authors

The authors have developed an algorithm called LEDA that is designed to separate the MS signals of structural isomers that do not yield single, specific fragment ions. This allows them to quantify individual isomers from mixtures without the need for prior chromatographic separation.

The authors have validated LEDA in previous publications using QqQ MS. Here, the authors apply the algorithm to ion trap MS.

The manuscript therefore describes the application of an existing technique to a different MS analyzer. Given the interest in isomer separations, this is a warranted goal. However, this raises the bar of what should be expected from the manuscript in terms of quality and understandability. As of yet, the manuscript is difficult to understand for readers unfamiliar with LEDA already. This is due to a combination of i) a need for English language editing; ii) many explanations required for understanding the results section (including the definition of abbreviations) are presented in the methods section; and iii) the figures not being furnished with legends (m/z, or otherwise).

The authors, following the reviewer’s suggestions, have accurately checked the English spelling, corrected the abbreviations and their definitions, added the legends requested in many figures. The answers to the reviewer’s remarks are stepwise reported below.

General remarks

Figures: The figures are provided in excellent quality/resolution. They would benefit greatly from legends explaining what EXACTLY is being shown (e.g. which m/z is plotted). Also, the concepts behind LEDA and the difference in mass analyzers used in this study (IT) compared to previous studies (QqQ) could be presented visually. This is hinted at in the comment to Fig. 2.

Self-citations: more than 39% of all references are self-citations (11/28). This is understandable, since LEDA was developed by the authors themselves, but nevertheless may raise concerns.

The authors believe that these references help the reader in the familiarization of LEDA and its features.

Specific comments

The title contains an uncommon abbreviation (LEDA).

LEDA, as reported in the introduction, is the acronym of Linear Equation of Deconvolution Analysis but can be considered as “tool” or "device" to elaborate the MS/MS data acquired during sample analysis. Furthermore, LEDA is the focus of the application and characterized the manuscript. Then the authors believe that LEDA, as all the devices or tool, should be present in the title.

34: MS is not necessarily "high-throughput." In many laboratories it is the bottleneck due to there being only one MS, while for many other instruments (e.g., HPLC-UV) multiple units are available.

The authors in this case describe the general characteristics of MS and the "high-throughput" is one of these (e.g. flow injection analysis, direct infusion, etc...). The instrumentation available in a common laboratory is not the subject of this sentence.

52: "Specificity" has a specific meaning in analytical chemistry. What is the meaning of the phrase "determination specificity" used here?

The authors meant as specificity “the ability to discriminate between compounds of closely related structures which are likely to be present in the sample”. ICH Q2B Guideline “Validation of Analytical Procedures Methodology” [ref. 27 in this manuscript].

60: Please specify what "large" in the phrase "relatively large collision gas (Argon)" refers to (diameter?).

The authors wanted to highlight the difference dimension between Helium and Argon used as collision gas in QqQ and IT mass analyzers respectively. Both are gas inert and monoatomic but they have very different shape (diameter or cross section) and atomic weight (4 vs 40 Da). These characteristics play a key role in the collision induced dissociation (CID) mechanisms and in the energization/excitation of the precursor ion. The intentions of the authors in the manuscript were to use the adjective "large" to include both the characteristics of Ar (shape and weight). However, to specify this topic, rephrase the sentence as follows: “All reported applications were performed, developed and evaluated by using a triple quadrupole system with conventional collision cell, collision induced dissociation (CID) mechanism and Argon as collision gas".

91: The abbreviation "MSn" needs to be defined (sequential MS).

The authors do not understand the reviewer remark, the acronym MSn was extracted from a sentence that define it: ".... reiteration of the tandem MS experiment on a product ion (MSn).

Throughout the manuscript, many abbreviations are used prior to their definition. An additional section ("abbreviations") might be helpful to collectively define the many abbreviations used throughout the manuscript (in addition to their definition within the text).

The authors followed the template, given by the journal to compile the manuscript, and the guidelines that reports: "Acronyms/Abbreviations/Initialisms should be defined the first time they appear in each of three sections: the abstract; the main text; the first figure or table. When defined for the first time, the acronym/abbreviation/initialism should be added in parentheses after the written-out form".

111: The term "product ions" is used twice, but once abbreviated (Pis) and once spelled out.

The reviewer extracted "product ions" in follows sentence: ".... acquires the MS/MS spectra in product ions scan, then all the Pis above the 5 % of abundance". Then, the term "product ions scan" is referred to the MS/MS method used, while "Pis" are referred all product ions formed during the MS/MS experiments.

Fig. 2: How could SY and PiF reach values greater than 100% as depicted by the fitted line? The denominator is the sum of two positive terms, one of them being the numerator.

The authors believe that this matter is not relevant since the experimental data were fitted by the software and the discrepancy is not meaningful.

Also, the concept of LEDA applied here to IT is currently difficult to understand for readers not familiar with its application to QqQ, i.e. readers unfamiliar with [22] or other manuscripts by the authors.

The authors, as reported above, included all the references in this manuscript can help the reader of the familiarization of LEDA and its features.

Hence, the following would be helpful additions (subfigures) to Fig. 2:

  1. A) A graphical visualization of the principle behind single-site fragmentation in ion trap MS contrasting "space" mass analyzer (QqQ) with "time" mass analyzer (IT);

The authors did not understand what the reviewer would like, large part of results and discussion sections of the manuscript are devoted to explain these topics.

  1. B) A visualization of what SY, PiF, and PiY correspond to (in addition to including a verbal explanation near Fig. 2, since the mathematical explanations are currently provided in the much later section 4.4).

The authors followed the guidelines of the journal that reports: “section Results, provide a concise and precise description of the experimental results, their interpretation as well as the experimental conclusions that can be drawn”. While the methods, protocol, mathematical formula, etc… were provided in M&M section.

  1. C) m/z or MS spectra of the observed SY, PiF, and PiY (e.g., SF12).

The authors did not understand what the reviewer would like, the figure SF12 shows the MS spectra of isomers, while the graph of SY, PiF, and PiY describes the variation of relative abundances (%) of the acquired signals (defined in section 4.4) in ERMS study.

Fig. 5: Please add the m/z of the precursor and product ion to emphasize them being the same for ELF94 and ELF96.

The authors corrected by following the reviewer suggestion.

193: Abbreviation "Ri" used but not defined until a later section. The same is true for other abbreviations such as PAR (line 268).

The authors corrected by following the reviewer suggestion.

Figure 6: Please label which chromatogram shows which compounds, as well as which peak corresponds to IS and which to ELF94/ELF96, so the reader does not have to be refered to ST1.

The authors corrected by following the reviewer suggestion.

213: What was the void time of the chromatographic system? This information is necessary to judge if the retention of IS and ELF94/ELF96 is sufficient to separate them from potential matrix components.

The authors measured the dead volume of chromatographic system at the beginning of the study and resulted 0.51 min. On the reviewer's requests this value was added in the caption of table ST1 as follows: "The void time measured for the chromatographic system used was 0.51 min".

Table 2: Even though it here refers to a validation plot of "expected vs. obtained" values, R2 by itself has been deprecated as sole measure of a calibration model's fit (e.g., Félix Camirand Lemyre et al., Squaring Things Up with R2: What It Is and What It Can (and Cannot) Tell You, Journal of Analytical Toxicology, https://doi.org/10.1093/jat/bkab036). Please add the accuracy and precision values obtained for each calibrator.

The authors complete agree with the reviewer, in fact the estimated linear function was evaluated both by determination coefficient or R2, that verify the proportionality between the variables, and with standard error of linear function or SE-Lin, that represents the standard deviation of the distances between the experimental points and the estimated linear function. The determination coefficient or R2 and the standard error of linear function or SE-Lin are reported in table 2. However, the authors proposed the use of validation plot only to evaluate the quantitative performance of LEDA on MS/MS analysis of mixtures of the analytes, it is not used as calibration curve. The "validation plot" was introduced to described with a single value the accuracy and/or the precision the quantitative ability of the LEDA elaboration in a range of concentration of the analytes, avoiding the list of the obtained values for each level of concentration tested (5 levels, 5 values of accuracy and 5 values of precision).

Please also add the accuracy and precision values for at least one set of QC controls.

The authors believe that this point has been explained in the previous sentences.

247: Given its detailed discussion (including the importance of the y-intercept near the origin), it may be helpful to show the validation plot in the manuscript and not in the supplementary information.

The introduction of the "validation plot" as evaluator of the LEDA quantitative performances was reported in the ref. 22, then its use cannot be considered as novelty. Furthermore, the manuscript includes eight figures and the authors do not believe necessary to insert another figure to describe the information given by the table 2. However, the reader can be found the figures of the validation plot obtained in this study in supplementary materials SF18-SF19.

Fig. 7: Why is the curve fitted only for ELF94 but not for ELF96? A slope near 1 would show its stability. Also, the labels given in the caption are incorrect (red circles instead of pink squares).

The authors corrected the caption following the reviewer suggestion.

Unfortunately, the slope that represents the plasma stability of ELF96 is close to zero (remember the y values are the natural logarithm of concentration found) and maintains a constant value during the experiment. Then, the linear regression cannot be applied, the variables are not correlated (R2 near to zero). Due to these evaluations, the authors decide to leave the experimental data points of the ELF96 unfitted, considering evident its stability behavior in tested human plasma samples.

Fig. 8: Where possible, please add the m/z legend for each trace to the figure to delineate single m/z traces from LEDA traces. This would be helpful also in previous chromatograms.

The authors corrected by following the reviewer suggestion (figures 4, 8, SF13 and SF20-23)

303: "did not work" might be re-phrased to not invoke the impression of the experiment having been unsuccessful. The fact that no discernible degradataion took place at time 0 was to be expected, and indeed is used here as a proof of concept.

The authors corrected by following the reviewer suggestion as follows: “The reconstructed profiles (Figure 8) showed an equal abundance of the isomers in the Ri signal, as of the mixture solution added, demonstrating that at time 0 plasma enzymes cannot hydrolyze the substrates”.

Section 4.3.: Were the final concentrations given in the text post-reconstitution? What was the final concentration of the IS?

The authors reporting the final concentration of each calibration level of the analytes in section 4.3 as follows: “Final concentrations of calibration levels of each analyte were: 5.0, 10.0, 25.0, 50.0, and 100.0 ng mL-1”. The IS is added in the same amount (0.3 mL of 33 ng mL-1) in each solution analyzed (calibrations, mixtures and samples). Since all the solution are diluted in 1 mL, the final concentration of IS is 10 ng mL-1.

Section 4.7.: What were the concentrations of each calibrator pre-sample preparation?

The authors reporting the preparation of standard solutions of analytes and IS in section 4.3 as follows: “Stock solutions of analytes and verapamil hydrochloride (internal standard or IS) were prepared in acetonitrile at 1.0 mg mL-1 and stored at 4 °C. Working solutions of each analyte were freshly prepared by diluting stock solutions up to a concentration of 1.0 µg mL-1 and 0.1 µg mL-1 (working solutions 1 and 2, respectively) in mixture of mQ water:acetonitrile 50:50 (v/v). The IS working solution was prepared in acetonitrile at 33 ng mL-1 (IS solution)”.

Round 2

Author Response

Dear Editor,

I received the second Reviewers’ comments to the manuscript "Isomers recognition in HPLC-MS/MS analysis of human plasma samples by using ion trap supported by LEDA data elaboration tool" and I wish to thank for their useful suggestions.

Enclosed please find the second revised version of the manuscript amended according to their suggestions.

A point-wise list of response to the comments and the corrections (red and blue text), is attached to this message.

The submitted manuscript has the full waiver publish charge provided by Mrs. YARA LIU and Mr. Beiner Zhang, granted me during the previous e-mail correspondences. I confirm that neither the manuscript nor any parts of its content are currently under consideration or published in another journal. All authors have approved the manuscript and agree with its submission to IJMS.

I hope that in present form the paper could be considered worthy of publication.

Thank you for the opportunity you gave us to resubmit the paper.

Waiting for again kind answer, I remain

Sincerely yours

Gianluca Bartolucci, (corresponding Author)

Response to Reviewer 1

General comments:

This paper describes a nice application of the analysis of co-eluted isomers by energy-resolved mass spectrometry. To this aim, the authors apply the LEDA algorithm. This algorithm has been applied by the authors before for the analysis of several isomeric compounds. The novelty, in this case, is that they show the feasibility of applying LEDA in the context of an ion trap instrument. All the applications described before were performed in QqQ instruments (widely used in chromatography) LEDA allows the correct quantification of both isomers. 

However, both isomers show specific transitions. ELF96 shows two specific fragments: 392 m/z and 253 m/z. And ELF94 shows also two specific transitions: 366 m/z and 279 m/z. 

The authors would like to elaborate the reviewer's sentence "However, both isomers show specific transitions". In present paper and in ref. 22 (QqQ study), the authors never report "specific transitions or specific product ions" but described the MS/MS spectra as formed by common product ions for both isomers with ".... different relative abundances". Therefore, each isomer, in the same MS/MS condition, produces the same pattern of product ions but with different branching ratios. Therefore, each isomer, in the same MS/MS condition, produces the same pattern of product ions but with different branching ratios. In our opinion, the reviewer's misunderstanding arises by the comparison between the MS/MS spectra of isomers (SF24-SF25) and/or the data of the characteristic ion abundance ratios reported in table 5. The "specific transitions" mentioned by the reviewer show a favored yield of formation (respect to the precursor ion) for an isomer over the other but does not exclude the mutual interference. How much can this interference affect the acquired signal? The case of the 392 m/z product ion that shows the yield of formation of 1% and 27 % (SD 1 %) for ELF94 and ELF96 respectively. But the isomers are simultaneously present in the sample and are co-eluted by the chromatographic column. Then, by taking a 90:10 mixture of isomers the 392 m/z signal is the sum of :

ELF94, 0.9 x 0.01 = 0.009;

ELF96, 0.1 x 0.27= 0.027;

Hence, the 25% (0.009/0.036) of the acquired 392 m/z signal are coming from the ELF94 isomer, that showing the lowest of yield of formation for this product ion.

From the MS/MS spectra showed in the supporting information is quite easy to misunderstand and think that ELF94 has no peaks at 392m/z and 253 m/z. The images have very good quality and even when I have amplified the images, I cannot see the peaks. And the same for ELF96: no peaks are seen for 366 m/z and 279 m/z. These fragments are even not reported in the MS/MS spectra.

The authors understand the reviewer's remarks on this topic and in previous answers have tried to explain the issue, but surely without the necessary clarity. Firstly, we have checked the figures SF24-SF25 and we identified the issue pointing out by reviewer; the MS/MS spectra were depicted with an abundance threshold major than 3%, then the low signals were hidden. Now, new SF24-SF25 figures were reported in supplementary materials with a proper threshold. The new figures SF24-SF25, reported below, show (green circles) the considered Pis are present, in different yield, in both isomers MS/MS spectra. Naturally, as remark the reviewer, the abundance of some Pis are very low (useless from quantitative point of view) but cannot be considered null.

The same can be understood from Table 5 of the manuscript. Ratios calculated as 0.01±0.01 (interval calculated as ±std) means that these ratios are not significantly different from 0. (the 95% confidence interval obtained as 2*std, assuming a normal distribution, would be then 0.01±0.02. Showing clearly that these coefficients are not significant).

The authors completely agree with the reviewer that the 95 % confidence interval is 2xSD, then for this case the relative abundance of considered Pi (e.g. 392 m/z) ranging between 0.03 to 0. In fact, the example reported in previous answer, the authors consider only the mean value (0.01), demonstrating that the 25% of the acquired 392 m/z signal in ELF94-ELF96 90:10 mixture are coming from the ELF94 isomer, by following the reviewer remark the interferences vary between 50% to 0%. However, they cannot be considered null or negligible.

I think that MRM could be applied in the present conditions. Maybe the sensitivity is higher if LEDA is applied (since the intensity of Ri is higher and, as showed in the new calibration curves of Supporting Information (SF26-SF30), the slope for Ri is higher than for the Pi 221 m/z, which corresponds to the product ion with highest intensity). So one could assume that if specific fragments (392 m/z and 366m/z) were used to quantify both isomers without LEDA, maybe lower LOD would be obtained.

The authors would like to remember that MRM is a typical acquisition mode of the QqQ, in IT the MS/MS signal is acquired in a m/z range (product ion scan), then having the sum of the abundances of all fragmented ions in the monitored m/z range. However, we can extract the signal of each selected Pi from the total ion signal and process it as quantitative signal. To better explain this topic, we are reported below in figures 1 and 2 the chromatographic profiles of the Pi signal considered in this study for each pure isomer. The profiles of 392 m/z and 253 m/z in the MS/MS analysis of the ELF94 (Figure 1) show a peak (green circles) in correspondence of the others Pis (as reported above, their abundances are low but not null).

Figure 1: MS/MS chromatographic profile of each Pi signal considered in this study for ELF94 isomer (you can see the Figure 1 in uploaded file docx).

The same consideration can be made for 366 m/z and 279 m/z in the ELF96 (Figure 2). Therefore, in isomers co-elution condition, the MS/MS analysis of mixture of isomers each Pi signal is the sum of the contribution of the isomers present.

 Figure 2: MS/MS chromatographic profile of each Pi signal considered in this study for ELF96 isomer (you can see the Figure 2 in uploaded file docx).

I think that MRM could be applied in the present conditions. Maybe the sensitivity is higher if LEDA is applied (since the intensity of Ri is higher and, as showed in the new calibration curves of Supporting Information (SF26-SF30), the slope for Ri is higher than for the Pi 221 m/z, which corresponds to the product ion with highest intensity). So one could assume that if specific fragments (392 m/z and 366m/z) were used to quantify both isomers without LEDA, maybe lower LOD would be obtained. 

The authors believe that this point has been explained in the previous sentences. However, regarding the quantitative elaboration and the use of Ri signal, the reason is also to have a similar sensitivity (slope values) between two isomers, condition that does not occur by using the most abundant Pi in MS/MS spectra of each isomer, due to the different formation yield.

This means that MRM could be applied without the need of using LEDA and deconvolution of the chromatograms. Extract Ion chromatograms could be obtained for both co-eluted isomers. Even, one of the fragments (392m/z) could be used as target ion and 253 m/z as qualitative ion (less intensity). The same for ELF94. 366 m/z could be the target ion and 279 m/z would be the qualitative ion. I think that the authors should compare the results obtained by LEDA with the ones of classical MRM (widely used in chromatography).

Joining with the previous answer, the application of the classical MRM procedure must provide the separation of the isomers (in different sample or by chromatography) to avoid mutual signal interferences.

I think that the interferences with relative intensities that are not significantly different from 0 are negligible even if the isomers are not chromatographically separated. 

The authors, to meet the reviewer's remarks, carried out the classical quantitative procedure by considering 366 m/z and 392 m/z as “signals specific” for ELF94 and ELF96 respectively. Then, we processed the mixtures 90:10 and 10:90 of the two isomers obtaining the results as follows:

Mix 90:10

ELF94 109 ng/mL ± 5 ng/mL (instead of 90 ng/mL);

ELF96 16 ng/mL ± 1 ng/mL (instead of 10 ng/mL);

Mix 10:90

ELF94 18 ng/mL ± 3 ng/mL (instead of 10 ng/mL);

ELF96 122 ng/mL ± 2 ng/mL (instead of 90 ng/mL);

The obtained results show an overestimation of quantitative data for both the isomers (20%-25%), demonstrating the mutual interferences among the Pis signals (as expected from example reported previously). Then, as the authors described in the manuscript, the MS/MS spectra are formed by common product ions for both isomers with ".... different relative abundances". Therefore, each isomer, in the same MS/MS condition, produces the same pattern of product ions but with different branching ratios and the MS/MS analysis of isomers mixture, in co-elution condition, each Pi signal is the sum of the contribution of the present isomers.

What is the added value of applying LEDA compared to classical MRM? 

The use of a methodological approach that simplifies the liquid chromatography (HPLC) parameters, allowing the use of a short column and a fast elution gradient, leading to an increased productivity without losing determination specificity. With this approach, the chromatographic column was used only to avoid or limit the interference of the sample matrix towards the analyte ionization process (matrix effects). Then, the MS/MS properties were explored to solve the signal assignment.

I think also that in this case (coelution of isomers) MRM could be applied. Another issue is that LEDA could give better quantification performances. This is why I was asking to compare MRM with LEDA. 

The authors believe that this point has been explained in the previous sentences.

I think that LEDA is necessary when there are not specific fragments. But in this case, there are specific fragments for both isomers. A comparison of the performance parameters obtained in MRM and LEDA for the analysis of both isomers should be shown in the manuscript. Moreover, it would be interesting to show the performances of LEDA in the case where there are no specific fragments. In this case, obtaining good quantitative results is more challenging than in the case showed. (from 6 fragments, 4 are specific). 

The authors believe that this point has been explained in the previous sentences.

It would be interesting also if the authors show what is the added value of using an Ion Trap instead of a QqQ. It is stated the disadvantage of Ion Trap: more time to measure than in QqQ and the necessity of separating the internal standard from the isomers in order to do two acquisition windows. However, what is the advantage of Ion Trap compared to QqQ? This maybe could be more clearly explained in the paper. To justify the use of ion trap instead of QqQ.

The authors have proposed the question: "Is it possible to transfer the MS/MS data post-processing LEDA approach to another type of tandem mass spectrometry system?". Then, the purpose is not finding "....the added value of using an Ion Trap instead of a QqQ" but if the transfer is feasible. The opinion of the authors that the aim was clearly explained in the paper (section 2: results and section 3: discussion). 

OK, I understand that the objective was to see the feasibility of using an ion trap instead of QqQ despite the limitations of the ion trap pointed by the authors.

Specific comments:

There are some mistakes to correct in English. Without being exhaustive, some mistakes to correct are:

  • Page 5, line 161. “are report” should be corrected by “are reported”

The authors corrected by following the reviewer suggestion.

  • Page 3, line 112. “A typical graphs used to describe…” should be maybe “a typical graph”

The authors corrected by following the reviewer suggestion.

  • Page 8, line 220. “the MS/MS data from the analyzes of the mixture”. It should be corrected “the MS/MS data from the analysis of the mixture”

The authors corrected by following the reviewer suggestion.

  • Page 13. Line 423 “The SY curve describe” should be “describes”

The authors corrected by following the reviewer suggestion.

  • Supporting Information. Legend ST1 “for each analyte obtained apply the proposed HPLC-MS/MS approach.” Correct maybe by “obtained with the proposed…”

The authors corrected by following the reviewer suggestion.

Validation of LEDA. Table 2 (page 8). And support info Figure SF19. For the isomer ELF96, the intercept is -0.03. Is there a negative constant bias in the quantification of this isomer of about 3%? Usually, significance tests of the slope and the intercept are performed in order to evaluate the absence of significant constant and proportional bias.

The authors proposed the use of validation plot only to evaluate the quantitative performance of LEDA, then it is not used as calibration curve. 

I know obviously that this regression line (expected values vs LEDA results) it is not a calibration curve used for quantification. This regression line is used to test if there is a proportional bias and a constant bias (as pointed by the authors in page 9: “All the slope values shown in Table 2 are close to 1” and “Then, together with the y-intercept values near 250 the origin, the assembled LEDA algorithm achieved a correct…” However, to validate a method usually hypothesis tests are done. In this case, a t-test for the slope and intercept. Or even, joint-confidence intervals to verify simultaneously that the slope is not significantly different from 1 and that the intercept is not significantly different from 0. (this information can be found for instance in Handbook of Chemometrics and Qualimetrics. Part A, Massart et al).

As suggested by the reviewer, for assessing the accuracy, that is the agreement between the expected values and the LEDA results, a regression slope t-test has been performed. The t-test was carried out at 95% confidence interval with n-2=3 degrees of freedom (calculated t= 0.412 for ELF96, critical t=3.182) and confirmed that the calculated value of the slope (0.9955) is not statistically different from 1. In the same way, a regression intercept t-test was carried out for verifying that the intercept is not statistically different from 0, with n-2=3 degrees of freedom (calculated t= 2.126 for ELF96, critical t=3.182). The calculated value of the intercept (-0.0308) was found not statistically different from 0, confirming that there was no negative systematic error in the determination. Data related to the tests were not added in the main text, but a brief comment was added in caption of figure SF19.

The LOD is calculated from the calibration curve (standard error divided by the slope). However, the human serum samples are diluted ten times for the chromatographic analysis. This means that the LOD and LOQ should be multiplied by 10, if the method is applied in serum samples.

The authors are conscious of reviewer's remark, but the proposed application does not assay the residues of studied analytes eventually present in unknown plasma samples. The samples were spiked with a known amount of analyte (1 µM), then the target of this study was the evaluation of the chemical stability of each analyte in experimental conditions (i.e. it maintains the initial concentration or decrease). Taking into the account of the aim of the study, the authors considered the parameters of quantitative determination (slope, intercept, LOD, etc..) of each analyte in the sample solution actually analyzed by the HPLC-MS/MS methods. Therefore, they believe that the reported values of LOD are better suited to represent the detection characteristics of the HPLC-MS/MS methods for this purpose.

In fact, to clarify this topic, the authors reported in the 4.7 section the sentence as follows: “Taking into the account that both calibration and sample solutions were prepared with the same procedure, the calibration curve parameters and the LOD value for each analyte were referred to the concentration of sample solution actually analyzed by the HPLC-MS/MS methods.” 

OK for the sentence. However, the active compound should be theoretically quantified from human plasma not diluted (so maybe, it could be clarified more clearly in the manuscript that if the method is to be applied to analyse serum samples in routine analysis, the LOD should be multiplied by 10 due to the dilution of the sample prior to analysis). 

As suggested by reviewer, the authors added to the section 4.6 Sample preparation the sentence as follows: “Following the proposed procedure, the human plasma matrix was diluted with a ratio 1:10 (i.e. 0.1 mL of human plasma at 1 mL final sample solution)”.

Internal standard calibration curve. Some explanations are given in section 4.7. and also in page 9: “Then, the isomers abundances were assigned by distributing them in the signal of precursor ion (or Ri signal), the corresponding PAR value calculated, by dividing with the IS abundance, and quantitative determination of the analytes in the sample performed by using the respective precursor ion calibration curve” And the results from the calibration curve are shown in supporting info (table ST2).

The authors are rephrased the sentence in page 9 as follows: “Then, the isomers abundances were assigned by distributing them in the signal of precursor ion (or Ri signal), the corresponding peak area ratio (PAR) value calculated, by dividing each isomer assigned peak area with the IS abundance, and quantitative determination of the analytes in the sample performed by using the respective precursor ion calibration curve”.

OK, Now with the new information included in the manuscript and supporting information it is easier to understand how the internal standard calibration curves have been done. A small comment maybe the legends SF26-SF30 could be start with “Internal standard calibration curve…” and also in the legend PAR could be again defined as the ratio between…

The authors agree with the reviewer and added in caption of figures SF26-SF29 as follows: “Internal standard calibration curve considering the peak area ratio (PAR) between Ri vs IS signals obtained by….”. The figure SF30 remains unchanged, in fact it represents the degradation plots of the reference compound (KEE) used to verify the hydrolytic activity of human plasma samples employed in this study.

I think the authors should explain more clearly the calculations. From the paragraph in page 9, it is possible to guess that the intensity plotted for each analyte corresponds probably to the precursor ion measured at excitation voltage of 20 V (before fragmentation of the precursor ion). But, what is the intensity measured for the internal standard? Table 4 (page 14) shows only that IS (455 m/z) was measured at 35 V and only the fragments were measured (between 135 and 350 m/z). The intensity of the IS corresponds to the total intensity of the fragments of IS? Or is there an error in Table 4, and also the IS was measured at 20 V to have the intensity of the precursor ion of IS? Moreover in supporting info, table ST2 shows two calibration curves for both isomers. One for Pi and the other one for Ri. What does it mean Pi, in this case? 

The authors agree with the reviewer suggestion and added a new table in supplementary materials (ST2) and in section 4.7 the following sentences: “The proposed IT-MS/MS method is assembled with three Pis scan events, each one characterized by a different m/z range acquisition and specific energetic conditions. Time segments, MS/MS parameters and the selected of ionic signals for monitoring the IS and each isomer for quantitative determination are reported in supplementary materials (ST2)”. “The comparison among the calibration curves demonstrates how the slope values of the Ri signals are similar for both isomers and higher respect to those obtained with monitoring of Pi1. This differences can be related to the characteristics of fragmentation of each isomer then, to maintain the same sensivity and quantitative relevance, the determination of the studied analytes was performed by using the peak abundances assigned by LEDA elaboration on Ri signal acquired from the sample [22]”. The old ST2 in supplementary materials, now is ST3.

The new information has improved the clarity of this part of the manuscript.

The name of the column is MRM transition. 

The authors renamed the column in ST3 as “MS/MS signal”

OK for the modification

So, one could think that Pi corresponds only to one transition? Is this the case? If yes, which transition? And if not, does it correspond to the sum of all the 6 selected product ions? I think that this part should be better explained. This is quite important, because the importance of this paper is focused on the quantification of the isomeric compounds. 

The authors believe that this point has been explained in the previous sentences.

OK

Also, a figure with the calibration curves in the supporting info could be included.

The authors, following the reviewer suggestion, added the figures SF26-SF29 of the calibration curves in supplementary materials.

OK

Reviewer 3 Report

The authors have implemented several changes leading to the improvement of the manuscript. However, not all issues have been addressed completely. I have highlighted the respective comments in yellow.

Comments and Suggestions for Authors

The authors have developed an algorithm called LEDA that is designed to separate the MS signals of structural isomers that do not yield single, specific fragment ions. This allows them to quantify individual isomers from mixtures without the need for prior chromatographic separation.

The authors have validated LEDA in previous publications using QqQ MS. Here, the authors apply the algorithm to ion trap MS.

The manuscript therefore describes the application of an existing technique to a different MS analyzer. Given the interest in isomer separations, this is a warranted goal. However, this raises the bar of what should be expected from the manuscript in terms of quality and understandability. As of yet, the manuscript is difficult to understand for readers unfamiliar with LEDA already. This is due to a combination of i) a need for English language editing; ii) many explanations required for understanding the results section (including the definition of abbreviations) are presented in the methods section; and iii) the figures not being furnished with legends (m/z, or otherwise).

The authors, following the reviewer’s suggestions, have accurately checked the English spelling, corrected the abbreviations and their definitions, added the legends requested in many figures. The answers to the reviewer’s remarks are stepwise reported below.

General remarks

Figures: The figures are provided in excellent quality/resolution. They would benefit greatly from legends explaining what EXACTLY is being shown (e.g. which m/z is plotted). Also, the concepts behind LEDA and the difference in mass analyzers used in this study (IT) compared to previous studies (QqQ) could be presented visually. This is hinted at in the comment to Fig. 2.

Self-citations: more than 39% of all references are self-citations (11/28). This is understandable, since LEDA was developed by the authors themselves, but nevertheless may raise concerns.

The authors believe that these references help the reader in the familiarization of LEDA and its features.

This is not a decision for a reviewer. I refer to the editor.

Specific comments

The title contains an uncommon abbreviation (LEDA).

LEDA, as reported in the introduction, is the acronym of Linear Equation of Deconvolution Analysis but can be considered as “tool” or "device" to elaborate the MS/MS data acquired during sample analysis. Furthermore, LEDA is the focus of the application and characterized the manuscript. Then the authors believe that LEDA, as all the devices or tool, should be present in the title.

I agree that the acronym should be part of the title, but not unexplained. I refer to the general convention of not using acronyms in titles unless they are not universally understood. Given that the authors cite exclusively themselves as users of LEDA (see comment above), LEDA does not yet appear to have reached mainstream usage.

34: MS is not necessarily "high-throughput." In many laboratories it is the bottleneck due to there being only one MS, while for many other instruments (e.g., HPLC-UV) multiple units are available.

The authors in this case describe the general characteristics of MS and the "high-throughput" is one of these (e.g. flow injection analysis, direct infusion, etc...). The instrumentation available in a common laboratory is not the subject of this sentence.

The issue has been resolved.

52: "Specificity" has a specific meaning in analytical chemistry. What is the meaning of the phrase "determination specificity" used here?

The authors meant as specificity “the ability to discriminate between compounds of closely related structures which are likely to be present in the sample”. ICH Q2B Guideline “Validation of Analytical Procedures Methodology” [ref. 27 in this manuscript].

What is the difference between “specificity” as defined by ICH and “determination specificity” as used in the manuscript?

Interfacing HPLC with MS generally leads to an increase in selectivity. Why do the authors note that a loss of “determination [?] specificity” was prevented if this was not be expected?

60: Please specify what "large" in the phrase "relatively large collision gas (Argon)" refers to (diameter?).

The authors wanted to highlight the difference dimension between Helium and Argon used as collision gas in QqQ and IT mass analyzers respectively. Both are gas inert and monoatomic but they have very different shape (diameter or cross section) and atomic weight (4 vs 40 Da). These characteristics play a key role in the collision induced dissociation (CID) mechanisms and in the energization/excitation of the precursor ion. The intentions of the authors in the manuscript were to use the adjective "large" to include both the characteristics of Ar (shape and weight). However, to specify this topic, rephrase the sentence as follows: “All reported applications were performed, developed and evaluated by using a triple quadrupole system with conventional collision cell, collision induced dissociation (CID) mechanism and Argon as collision gas".

Thank you for the explanation. The issue has been resolved.

91: The abbreviation "MSn" needs to be defined (sequential MS).

The authors do not understand the reviewer remark, the acronym MSn was extracted from a sentence that define it: ".... reiteration of the tandem MS experiment on a product ion (MSn).

In the old version of the manuscript, the abbreviation used was “MSn” (without superscript) while the newly submitted manuscript shows “MSn” (with superscript). It is apparent now that the n refers to the iteration of MS. The issue has been resolved.

Throughout the manuscript, many abbreviations are used prior to their definition. An additional section ("abbreviations") might be helpful to collectively define the many abbreviations used throughout the manuscript (in addition to their definition within the text).

The authors followed the template, given by the journal to compile the manuscript, and the guidelines that reports: "Acronyms/Abbreviations/Initialisms should be defined the first time they appear in each of three sections: the abstract; the main text; the first figure or table. When defined for the first time, the acronym/abbreviation/initialism should be added in parentheses after the written-out form".

I do not take issue with this. None the less, an additional section explaining all acronyms would be of help to the reader.

111: The term "product ions" is used twice, but once abbreviated (Pis) and once spelled out.

The reviewer extracted "product ions" in follows sentence: ".... acquires the MS/MS spectra in product ions scan, then all the Pis above the 5 % of abundance". Then, the term "product ions scan" is referred to the MS/MS method used, while "Pis" are referred all product ions formed during the MS/MS experiments.

I do not fully understand the reply:
What do you mean by “the term … is referred to the MS/MS method used”?
Do you mean “the term … refers to the MS/MS method used”?

More importantly, the authors’ extract is mistaken. Here is line 111 in the original manuscript, which I referred to in my comment:

“The evaluation of these experiments was carried out by plotting the survival yield curves of precursor ion (SY), the Pis formation (PiF) and the product ions yield (PiY, see details in section 4.4).”

Both PiF and PiY refer to “product ions” which can be abbreviated as “Pis”.

Once it is abbreviated (“Pis formation”).
Once it is spelled out (“product ions yield”).
This is not a big problem; it is just inconsistent.

Fig. 2: How could SY and PiF reach values greater than 100% as depicted by the fitted line? The denominator is the sum of two positive terms, one of them being the numerator.

The authors believe that this matter is not relevant since the experimental data were fitted by the software and the discrepancy is not meaningful.

I cannot judge the relevancy of this. I just pointed out the practical impossibility of the presented data, which A) will also surprise future readers and B) may suggest using a different fitting model.

Also, the concept of LEDA applied here to IT is currently difficult to understand for readers not familiar with its application to QqQ, i.e. readers unfamiliar with [22] or other manuscripts by the authors.

The authors, as reported above, included all the references in this manuscript can help the reader of the familiarization of LEDA and its features.

Hence, the following would be helpful additions (subfigures) to Fig. 2:

A) A graphical visualization of the principle behind single-site fragmentation in ion trap MS contrasting "space" mass analyzer (QqQ) with "time" mass analyzer (IT);

The authors did not understand what the reviewer would like, large part of results and discussion sections of the manuscript are devoted to explain these topics.

B) A visualization of what SY, PiF, and PiY correspond to (in addition to including a verbal explanation near Fig. 2, since the mathematical explanations are currently provided in the much later section 4.4).

The authors followed the guidelines of the journal that reports: “section Results, provide a concise and precise description of the experimental results, their interpretation as well as the experimental conclusions that can be drawn”. While the methods, protocol, mathematical formula, etc… were provided in M&M section.

C) m/z or MS spectra of the observed SY, PiF, and PiY (e.g., SF12).

The authors did not understand what the reviewer would like, the figure SF12 shows the MS spectra of isomers, while the graph of SY, PiF, and PiY describes the variation of relative abundances (%) of the acquired signals (defined in section 4.4) in ERMS study.

I refer this issue (A-C) to the editor. As it stands, the manuscript provides little novelty while requiring readers to be extensively familiar with the previous publications by the authors. This could be offset by providing the clearest possible explanation of the (re-)applied methodology, preferably with the help of explanatory figures. This may be of particular relevancy when considering that the journal the manuscript was submitted to does not specialize in separation science, let alone MS computational algorithms.

Fig. 5: Please add the m/z of the precursor and product ion to emphasize them being the same for ELF94 and ELF96.

The authors corrected by following the reviewer suggestion.

The issue has been resolved.

193: Abbreviation "Ri" used but not defined until a later section. The same is true for other abbreviations such as PAR (line 268).

The authors corrected by following the reviewer suggestion.

The issue has been resolved.

Figure 6: Please label which chromatogram shows which compounds, as well as which peak corresponds to IS and which to ELF94/ELF96, so the reader does not have to be refered to ST1.

The authors corrected by following the reviewer suggestion.

The issue has been resolved.

213: What was the void time of the chromatographic system? This information is necessary to judge if the retention of IS and ELF94/ELF96 is sufficient to separate them from potential matrix components.

The authors measured the dead volume of chromatographic system at the beginning of the study and resulted 0.51 min. On the reviewer's requests this value was added in the caption of table ST1 as follows: "The void time measured for the chromatographic system used was 0.51 min".

This translates to sufficiently high k values to expect an effective removal of matrix components. The issue has been resolved.

Table 2: Even though it here refers to a validation plot of "expected vs. obtained" values, R2 by itself has been deprecated as sole measure of a calibration model's fit (e.g., Félix Camirand Lemyre et al., Squaring Things Up with R2: What It Is and What It Can (and Cannot) Tell You, Journal of Analytical Toxicology, https://doi.org/10.1093/jat/bkab036). Please add the accuracy and precision values obtained for each calibrator.

The authors complete agree with the reviewer, in fact the estimated linear function was evaluated both by determination coefficient or R2, that verify the proportionality between the variables, and with standard error of linear function or SE-Lin, that represents the standard deviation of the distances between the experimental points and the estimated linear function. The determination coefficient or R2 and the standard error of linear function or SE-Lin are reported in table 2. However, the authors proposed the use of validation plot only to evaluate the quantitative performance of LEDA on MS/MS analysis of mixtures of the analytes, it is not used as calibration curve. The "validation plot" was introduced to described with a single value the accuracy and/or the precision the quantitative ability of the LEDA elaboration in a range of concentration of the analytes, avoiding the list of the obtained values for each level of concentration tested (5 levels, 5 values of accuracy and 5 values of precision).

The issue has been resolved.

Please also add the accuracy and precision values for at least one set of QC controls.

The authors believe that this point has been explained in the previous sentences.

The issue has been resolved.

247: Given its detailed discussion (including the importance of the y-intercept near the origin), it may be helpful to show the validation plot in the manuscript and not in the supplementary information.

The introduction of the "validation plot" as evaluator of the LEDA quantitative performances was reported in the ref. 22, then its use cannot be considered as novelty. Furthermore, the manuscript includes eight figures and the authors do not believe necessary to insert another figure to describe the information given by the table 2. However, the reader can be found the figures of the validation plot obtained in this study in supplementary materials SF18-SF19.

The issue has been resolved.

Fig. 7: Why is the curve fitted only for ELF94 but not for ELF96? A slope near 1 would show its stability. Also, the labels given in the caption are incorrect (red circles instead of pink squares).

The authors corrected the caption following the reviewer suggestion.

Unfortunately, the slope that represents the plasma stability of ELF96 is close to zero (remember the y values are the natural logarithm of concentration found) and maintains a constant value during the experiment. Then, the linear regression cannot be applied, the variables are not correlated (R2 near to zero). Due to these evaluations, the authors decide to leave the experimental data points of the ELF96 unfitted, considering evident its stability behavior in tested human plasma samples.

The issue has been resolved.

Fig. 8: Where possible, please add the m/z legend for each trace to the figure to delineate single m/z traces from LEDA traces. This would be helpful also in previous chromatograms.

The authors corrected by following the reviewer suggestion (figures 4, 8, SF13 and SF20-23)

The issue has been resolved.

303: "did not work" might be re-phrased to not invoke the impression of the experiment having been unsuccessful. The fact that no discernible degradataion took place at time 0 was to be expected, and indeed is used here as a proof of concept.

The authors corrected by following the reviewer suggestion as follows: “The reconstructed profiles (Figure 8) showed an equal abundance of the isomers in the Ri signal, as of the mixture solution added, demonstrating that at time 0 plasma enzymes cannot hydrolyze the substrates”.

The issue/comment has been resolved.

Section 4.3.: Were the final concentrations given in the text post-reconstitution? What was the final concentration of the IS?

The authors reporting the final concentration of each calibration level of the analytes in section 4.3 as follows: “Final concentrations of calibration levels of each analyte were: 5.0, 10.0, 25.0, 50.0, and 100.0 ng mL-1”. The IS is added in the same amount (0.3 mL of 33 ng mL-1) in each solution analyzed (calibrations, mixtures and samples). Since all the solution are diluted in 1 mL, the final concentration of IS is 10 ng mL-1.

The issue/comment has been resolved.

Section 4.7.: What were the concentrations of each calibrator pre-sample preparation?

The authors reporting the preparation of standard solutions of analytes and IS in section 4.3 as follows: “Stock solutions of analytes and verapamil hydrochloride (internal standard or IS) were prepared in acetonitrile at 1.0 mg mL-1 and stored at 4 °C. Working solutions of each analyte were freshly prepared by diluting stock solutions up to a concentration of 1.0 µg mL-1 and 0.1 µg mL-1 (working solutions 1 and 2, respectively) in mixture of mQ water:acetonitrile 50:50 (v/v). The IS working solution was prepared in acetonitrile at 33 ng mL-1 (IS solution)”.

The issue/comment has been resolved.

Author Response

Dear Editor,

I received the second Reviewers’ comments to the manuscript "Isomers recognition in HPLC-MS/MS analysis of human plasma samples by using ion trap supported by LEDA data elaboration tool" and I wish to thank for their useful suggestions.

Enclosed please find the second revised version of the manuscript amended according to their suggestions.

A point-wise list of response to the comments and the corrections (red and blue text), is attached to this message.

The submitted manuscript has the full waiver publish charge provided by Mrs. YARA LIU and Mr. Beiner Zhang, granted me during the previous e-mail correspondences. I confirm that neither the manuscript nor any parts of its content are currently under consideration or published in another journal. All authors have approved the manuscript and agree with its submission to IJMS.

I hope that in present form the paper could be considered worthy of publication.

Thank you for the opportunity you gave us to resubmit the paper.

Waiting for again kind answer, I remain

Sincerely yours

Gianluca Bartolucci, (corresponding Author)

Response to Reviewer 3

Comments and Suggestions for Authors

The authors have implemented several changes leading to the improvement of the manuscript. However, not all issues have been addressed completely. I have highlighted the respective comments in yellow.

Comments and Suggestions for Authors

The authors have developed an algorithm called LEDA that is designed to separate the MS signals of structural isomers that do not yield single, specific fragment ions. This allows them to quantify individual isomers from mixtures without the need for prior chromatographic separation.

The authors have validated LEDA in previous publications using QqQ MS. Here, the authors apply the algorithm to ion trap MS.

The manuscript therefore describes the application of an existing technique to a different MS analyzer. Given the interest in isomer separations, this is a warranted goal. However, this raises the bar of what should be expected from the manuscript in terms of quality and understandability. As of yet, the manuscript is difficult to understand for readers unfamiliar with LEDA already. This is due to a combination of i) a need for English language editing; ii) many explanations required for understanding the results section (including the definition of abbreviations) are presented in the methods section; and iii) the figures not being furnished with legends (m/z, or otherwise).

The authors, following the reviewer’s suggestions, have accurately checked the English spelling, corrected the abbreviations and their definitions, added the legends requested in many figures. The answers to the reviewer’s remarks are stepwise reported below.

General remarks

Figures: The figures are provided in excellent quality/resolution. They would benefit greatly from legends explaining what EXACTLY is being shown (e.g. which m/z is plotted). Also, the concepts behind LEDA and the difference in mass analyzers used in this study (IT) compared to previous studies (QqQ) could be presented visually. This is hinted at in the comment to Fig. 2.

Self-citations: more than 39% of all references are self-citations (11/28). This is understandable, since LEDA was developed by the authors themselves, but nevertheless may raise concerns.

The authors believe that these references help the reader in the familiarization of LEDA and its features.

This is not a decision for a reviewer. I refer to the editor.

The authors believe that this point has been explained in the previous answer.

Specific comments

The title contains an uncommon abbreviation (LEDA).

LEDA, as reported in the introduction, is the acronym of Linear Equation of Deconvolution Analysis but can be considered as “tool” or "device" to elaborate the MS/MS data acquired during sample analysis. Furthermore, LEDA is the focus of the application and characterized the manuscript. Then the authors believe that LEDA, as all the devices or tool, should be present in the title.

I agree that the acronym should be part of the title, but not unexplained. I refer to the general convention of not using acronyms in titles unless they are not universally understood. Given that the authors cite exclusively themselves as users of LEDA (see comment above), LEDA does not yet appear to have reached mainstream usage.

The authors following the reviewer’s suggestion removed the LEDA acronym from the title of the manuscript rephrase: “Isomers recognition in HPLC-MS/MS analysis of human plasma samples by using ion trap supported by a linear equations-based algorithm”.

34: MS is not necessarily "high-throughput." In many laboratories it is the bottleneck due to there being only one MS, while for many other instruments (e.g., HPLC-UV) multiple units are available.

The authors in this case describe the general characteristics of MS and the "high-throughput" is one of these (e.g. flow injection analysis, direct infusion, etc...). The instrumentation available in a common laboratory is not the subject of this sentence.

The issue has been resolved.

52: "Specificity" has a specific meaning in analytical chemistry. What is the meaning of the phrase "determination specificity" used here?

The authors meant as specificity “the ability to discriminate between compounds of closely related structures which are likely to be present in the sample”. ICH Q2B Guideline “Validation of Analytical Procedures Methodology” [ref. 27 in this manuscript].

What is the difference between “specificity” as defined by ICH and “determination specificity” as used in the manuscript?

Interfacing HPLC with MS generally leads to an increase in selectivity. Why do the authors note that a loss of “determination [?] specificity” was prevented if this was not be expected?

The authors, in many parts of the manuscript, point out just this topic: the proposed chromatography conditions do not guarantee the separation/distinction of the isomers (specificity). Then, to monitor the variation of the concentration of each isomer in the sample, we use the features of the MS/MS experiments supported by the LEDA algorithm. The authors proposed the approach as: a sole chromatographic condition that do not worries to separate the analytes, also if isomers, but allows fast, robust and reliable analysis of the samples. Therefore, when the analytes are isomers and detected in co-elution conditions, the authors must guarantee the identification of each analyte and its quantification (determination specificity). However, to meet the reviewer’s suggestion, we rephrased as follows: "....leading to higher productivity without losing the specificity of their determination".

60: Please specify what "large" in the phrase "relatively large collision gas (Argon)" refers to (diameter?).

The authors wanted to highlight the difference dimension between Helium and Argon used as collision gas in QqQ and IT mass analyzers respectively. Both are gas inert and monoatomic but they have very different shape (diameter or cross section) and atomic weight (4 vs 40 Da). These characteristics play a key role in the collision induced dissociation (CID) mechanisms and in the energization/excitation of the precursor ion. The intentions of the authors in the manuscript were to use the adjective "large" to include both the characteristics of Ar (shape and weight). However, to specify this topic, rephrase the sentence as follows: “All reported applications were performed, developed and evaluated by using a triple quadrupole system with conventional collision cell, collision induced dissociation (CID) mechanism and Argon as collision gas".

Thank you for the explanation. The issue has been resolved.

91: The abbreviation "MSn" needs to be defined (sequential MS).

The authors do not understand the reviewer remark, the acronym MSn was extracted from a sentence that define it: ".... reiteration of the tandem MS experiment on a product ion (MSn).

In the old version of the manuscript, the abbreviation used was “MSn” (without superscript) while the newly submitted manuscript shows “MSn” (with superscript). It is apparent now that the n refers to the iteration of MS. The issue has been resolved.

Throughout the manuscript, many abbreviations are used prior to their definition. An additional section ("abbreviations") might be helpful to collectively define the many abbreviations used throughout the manuscript (in addition to their definition within the text).

The authors followed the template, given by the journal to compile the manuscript, and the guidelines that reports: "Acronyms/Abbreviations/Initialisms should be defined the first time they appear in each of three sections: the abstract; the main text; the first figure or table. When defined for the first time, the acronym/abbreviation/initialism should be added in parentheses after the written-out form".

I do not take issue with this. None the less, an additional section explaining all acronyms would be of help to the reader.

The authors believe that this point has been explained in the previous answer.

111: The term "product ions" is used twice, but once abbreviated (Pis) and once spelled out.

The reviewer extracted "product ions" in follows sentence: ".... acquires the MS/MS spectra in product ions scan, then all the Pis above the 5 % of abundance". Then, the term "product ions scan" is referred to the MS/MS method used, while "Pis" are referred all product ions formed during the MS/MS experiments.

I do not fully understand the reply:
What do you mean by “the term … is referred to the MS/MS method used”?
Do you mean “the term … refers to the MS/MS method used”?

More importantly, the authors’ extract is mistaken. Here is line 111 in the original manuscript, which I referred to in my comment:

“The evaluation of these experiments was carried out by plotting the survival yield curves of precursor ion (SY), the Pis formation (PiF) and the product ions yield (PiY, see details in section 4.4).”

Both PiF and PiY refer to “product ions” which can be abbreviated as “Pis”.

Once it is abbreviated (“Pis formation”).
Once it is spelled out (“product ions yield”).
This is not a big problem; it is just inconsistent.

The authors extracted the sentence from the wrong line in the previous answer. Considering the line referred in the comment, the authors corrected the sentence as it follows: “The evaluation of these experiments was carried out by plotting the survival yield curves of precursor ion (SY), the Pis formation (PiF) and the Pis yield (PiY, see details in section 4.4)..”, as pointed out by the reviewer.

Fig. 2: How could SY and PiF reach values greater than 100% as depicted by the fitted line? The denominator is the sum of two positive terms, one of them being the numerator.

The authors believe that this matter is not relevant since the experimental data were fitted by the software and the discrepancy is not meaningful.

I cannot judge the relevancy of this. I just pointed out the practical impossibility of the presented data, which A) will also surprise future readers and B) may suggest using a different fitting model.

The authors following the reviewer’s suggestion used a different fitting model.

Also, the concept of LEDA applied here to IT is currently difficult to understand for readers not familiar with its application to QqQ, i.e. readers unfamiliar with [22] or other manuscripts by the authors.

The authors, as reported above, included all the references in this manuscript can help the reader of the familiarization of LEDA and its features.

Hence, the following would be helpful additions (subfigures) to Fig. 2:

  1. A) A graphical visualization of the principle behind single-site fragmentation in ion trap MS contrasting "space" mass analyzer (QqQ) with "time" mass analyzer (IT);

The authors did not understand what the reviewer would like, large part of results and discussion sections of the manuscript are devoted to explain these topics.

The authors believe that a graphical visualization of the ion trap fragmentation does not add information respect to the sections (results and discussion) already included in the manuscript, to meet the reviewer suggestion’s, we add three refs [25-27] that explain in detail this topic.

  1. B) A visualization of what SY, PiF, and PiY correspond to (in addition to including a verbal explanation near Fig. 2, since the mathematical explanations are currently provided in the much later section 4.4).

The authors followed the guidelines of the journal that reports: “section Results, provide a concise and precise description of the experimental results, their interpretation as well as the experimental conclusions that can be drawn”. While the methods, protocol, mathematical formula, etc… were provided in M&M section.

The authors believe that this point has been explained in the previous answer.

  1. C) m/z or MS spectra of the observed SY, PiF, and PiY (e.g., SF12).

The authors did not understand what the reviewer would like, the figure SF12 shows the MS spectra of isomers, while the graph of SY, PiF, and PiY describes the variation of relative abundances (%) of the acquired signals (defined in section 4.4) in ERMS study.

The authors believe that this point has been explained in the previous answer.

I refer this issue (A-C) to the editor. As it stands, the manuscript provides little novelty while requiring readers to be extensively familiar with the previous publications by the authors. This could be offset by providing the clearest possible explanation of the (re-)applied methodology, preferably with the help of explanatory figures. This may be of particular relevancy when considering that the journal the manuscript was submitted to does not specialize in separation science, let alone MS computational algorithms.

The authors are surprised and in the same time confused, the reviewer report "Given the interest in isomer separations, this is a warranted goal" , but in this remark, addressed to the editors, he claims "As it stands, the manuscript provides little novelty...." . The target of our investigations is to propose a different analytical approach in the isomers recognition by development of specific MS/MS methods, without support of the chromatographic resolution. The purpose is achieved step by step by identifying the problems, proposing solutions and verifying on a real application. In previous works, we demonstrated that specificity of the MS/MS methods can be achieved by playing on the energetic dimension of the experiment, but this is strongly linked by the MS analyzer employed. As reported in many sections of manuscript, the MS analyzer used is an ion trap (IT) which, due to its intrinsic characteristic, constitutes a challenge to refine the parameter combinations and find the right ion energization into MS/MS process, satisfying the specificity required for isomer recognition. Then, the submitted manuscript detailed reports the encountered issues, the differences with previous MS/MS experiments (carried out with a triple quadrupole), the introduction of new evaluation parameters to check the energetic dimension of IT MS/MS mechanism, the verification of the optimized conditions on a known real application. Reading the proposed manuscript, the main novelties are located pointwise:

- introduction of a different MS analyzer (IT);

- different MS/MS spectra obtained and different processing procedure introduced;

- overcome of the different energetic management of IT;

- MS/MS optimized conditions are assembled and fitted to monitor the sample chromatographic profile.

Each of these novelties are reported, described and verified. The only similarity with the previous work is in the application used to verify the reliability of the proposed IT MS/MS method, which is needed to compare the obtained results. In our opinion all the information reported in the manuscript are worthy to be published and could be useful to share with the readers of the journal. The authors believe that the sharing of knowledge, experiences and new approaches should not be limited in dedicated journal but it should be disseminated as widely as possible. On the other hand, the manuscript is a report on the results obtained by the authors in the topic described in the title, we do not consider useful "explanatory figures" on operating principles of QqQ and/or IT, while we think that the introduction of our experiences could be a reason for further study by readers, perhaps by following the references provided.

Fig. 5: Please add the m/z of the precursor and product ion to emphasize them being the same for ELF94 and ELF96.

The authors corrected by following the reviewer suggestion.

The issue has been resolved.

193: Abbreviation "Ri" used but not defined until a later section. The same is true for other abbreviations such as PAR (line 268).

The authors corrected by following the reviewer suggestion.

The issue has been resolved.

Figure 6: Please label which chromatogram shows which compounds, as well as which peak corresponds to IS and which to ELF94/ELF96, so the reader does not have to be refered to ST1.

The authors corrected by following the reviewer suggestion.

The issue has been resolved.

213: What was the void time of the chromatographic system? This information is necessary to judge if the retention of IS and ELF94/ELF96 is sufficient to separate them from potential matrix components.

The authors measured the dead volume of chromatographic system at the beginning of the study and resulted 0.51 min. On the reviewer's requests this value was added in the caption of table ST1 as follows: "The void time measured for the chromatographic system used was 0.51 min".

This translates to sufficiently high k values to expect an effective removal of matrix components. The issue has been resolved.

Table 2: Even though it here refers to a validation plot of "expected vs. obtained" values, R2 by itself has been deprecated as sole measure of a calibration model's fit (e.g., Félix Camirand Lemyre et al., Squaring Things Up with R2: What It Is and What It Can (and Cannot) Tell You, Journal of Analytical Toxicology, https://doi.org/10.1093/jat/bkab036). Please add the accuracy and precision values obtained for each calibrator.

The authors complete agree with the reviewer, in fact the estimated linear function was evaluated both by determination coefficient or R2, that verify the proportionality between the variables, and with standard error of linear function or SE-Lin, that represents the standard deviation of the distances between the experimental points and the estimated linear function. The determination coefficient or R2 and the standard error of linear function or SE-Lin are reported in table 2. However, the authors proposed the use of validation plot only to evaluate the quantitative performance of LEDA on MS/MS analysis of mixtures of the analytes, it is not used as calibration curve. The "validation plot" was introduced to described with a single value the accuracy and/or the precision the quantitative ability of the LEDA elaboration in a range of concentration of the analytes, avoiding the list of the obtained values for each level of concentration tested (5 levels, 5 values of accuracy and 5 values of precision).

The issue has been resolved.

Please also add the accuracy and precision values for at least one set of QC controls.

The authors believe that this point has been explained in the previous sentences.

The issue has been resolved.

247: Given its detailed discussion (including the importance of the y-intercept near the origin), it may be helpful to show the validation plot in the manuscript and not in the supplementary information.

The introduction of the "validation plot" as evaluator of the LEDA quantitative performances was reported in the ref. 22, then its use cannot be considered as novelty. Furthermore, the manuscript includes eight figures and the authors do not believe necessary to insert another figure to describe the information given by the table 2. However, the reader can be found the figures of the validation plot obtained in this study in supplementary materials SF18-SF19.

The issue has been resolved.

Fig. 7: Why is the curve fitted only for ELF94 but not for ELF96? A slope near 1 would show its stability. Also, the labels given in the caption are incorrect (red circles instead of pink squares).

The authors corrected the caption following the reviewer suggestion.

Unfortunately, the slope that represents the plasma stability of ELF96 is close to zero (remember the y values are the natural logarithm of concentration found) and maintains a constant value during the experiment. Then, the linear regression cannot be applied, the variables are not correlated (R2 near to zero). Due to these evaluations, the authors decide to leave the experimental data points of the ELF96 unfitted, considering evident its stability behavior in tested human plasma samples.

The issue has been resolved.

Fig. 8: Where possible, please add the m/z legend for each trace to the figure to delineate single m/z traces from LEDA traces. This would be helpful also in previous chromatograms.

The authors corrected by following the reviewer suggestion (figures 4, 8, SF13 and SF20-23)

The issue has been resolved.

303: "did not work" might be re-phrased to not invoke the impression of the experiment having been unsuccessful. The fact that no discernible degradataion took place at time 0 was to be expected, and indeed is used here as a proof of concept.

The authors corrected by following the reviewer suggestion as follows: “The reconstructed profiles (Figure 8) showed an equal abundance of the isomers in the Ri signal, as of the mixture solution added, demonstrating that at time 0 plasma enzymes cannot hydrolyze the substrates”.

The issue/comment has been resolved.

Section 4.3.: Were the final concentrations given in the text post-reconstitution? What was the final concentration of the IS?

The authors reporting the final concentration of each calibration level of the analytes in section 4.3 as follows: “Final concentrations of calibration levels of each analyte were: 5.0, 10.0, 25.0, 50.0, and 100.0 ng mL-1”. The IS is added in the same amount (0.3 mL of 33 ng mL-1) in each solution analyzed (calibrations, mixtures and samples). Since all the solution are diluted in 1 mL, the final concentration of IS is 10 ng mL-1.

The issue/comment has been resolved.

Section 4.7.: What were the concentrations of each calibrator pre-sample preparation?

The authors reporting the preparation of standard solutions of analytes and IS in section 4.3 as follows: “Stock solutions of analytes and verapamil hydrochloride (internal standard or IS) were prepared in acetonitrile at 1.0 mg mL-1 and stored at 4 °C. Working solutions of each analyte were freshly prepared by diluting stock solutions up to a concentration of 1.0 µg mL-1 and 0.1 µg mL-1 (working solutions 1 and 2, respectively) in mixture of mQ water:acetonitrile 50:50 (v/v). The IS working solution was prepared in acetonitrile at 33 ng mL-1 (IS solution)”.

The issue/comment has been resolved.

Round 3

Reviewer 3 Report

The authors' have resolved the majority of issues.

I refer the unresolved issues (summarized in the enclosed PDF) to the editor. 

Author Response

Dear Editors,

I received the Reviewers’ comments to the manuscript for third time "Isomers recognition in HPLC-MS/MS analysis of human plasma samples by using ion trap supported by LEDA data elaboration tool".

Enclosed please find the fourth revised version of the manuscript amended according to their suggestions.

A point-wise list of response to the comments and the corrections (red, blue and purple text), is attached to this message.

I confirm that neither the manuscript nor any parts of its content are currently under consideration or published in another journal. All authors have approved the manuscript and agree with its submission to IJMS.

I hope that in present form the paper could be considered worthy of publication.

Thank you for the opportunity you gave us to resubmit the paper.

Waiting for again kind answer, I remain

Sincerely yours

Gianluca Bartolucci, (corresponding Author)

Comment 1:

Fig. 2: How could SY and PiF reach values greater than 100% as depicted by the fitted line? The denominator is the sum of two positive terms, one of them being the numerator.

The authors believe that this matter is not relevant since the experimental data were fitted by the software and the discrepancy is not meaningful.

I cannot judge the relevancy of this. I just pointed out the practical impossibility of the presented data, which A) will also surprise future readers and B) may suggest using a different fitting model.

The authors following the reviewer’s suggestion used a different fitting model.

The comment referred to values greater than 100% as depicted by the fitted line. The authors claim to have responded by using a different fitting model. Shown below are the two versions of Figure 2 with the “changed” fitting model on the left (v3) and the original figure on the right (v1). They are identical.

The authors apologize for the misunderstanding, they believed that the reviewer's remarks concerned other similar graphs (among SF1-SF9) that showed more obvious fitting deviations. Actually, the figure 2 was not corrected. However, a new figure 2 is now reported in the manuscript with the proper fitting (as reported bottom).

Comment 2:

I refer this issue (A-C) to the editor. As it stands, the manuscript provides little novelty while requiring readers to be extensively familiar with the previous publications by the authors. This could be offset by providing the clearest possible explanation of the (re-)applied methodology, preferably with the help of explanatory figures. This may be of particular relevancy when considering that the journal the manuscript was submitted to does not specialize in separation science, let alone MS computational algorithms.

The authors are surprised and in the same time confused, the reviewer report "Given the interest in isomer separations, this is a warranted goal" , but in this remark, addressed to the editors, he claims "As it stands, the manuscript provides little novelty...." . The target of our investigations is to propose a different analytical approach in the isomers recognition by development of specific MS/MS methods, without support of the chromatographic resolution. The purpose is achieved step by step by identifying the problems, proposing solutions and verifying on a real application. In previous works, we demonstrated that specificity of the MS/MS methods can be achieved by playing on the energetic dimension of the experiment, but this is strongly linked by the MS analyzer employed. As reported in many sections of manuscript, the MS analyzer used is an ion trap (IT) which, due to its intrinsic characteristic, constitutes a challenge to refine the parameter combinations and find the right ion energization into MS/MS process, satisfying the specificity required for isomer recognition. Then, the submitted manuscript detailed reports the encountered issues, the differences with previous MS/MS experiments (carried out with a triple quadrupole), the introduction of new evaluation parameters to check the energetic dimension of IT MS/MS mechanism, the verification of the optimized conditions on a known real application. Reading the proposed manuscript, the main novelties are located pointwise:

- introduction of a different MS analyzer (IT);

- different MS/MS spectra obtained and different processing procedure introduced;

- overcome of the different energetic management of IT;

- MS/MS optimized conditions are assembled and fitted to monitor the sample chromatographic profile.

Each of these novelties are reported, described and verified. The only similarity with the previous work is in the application used to verify the reliability of the proposed IT MS/MS method, which is needed to compare the obtained results. In our opinion all the information reported in the manuscript are worthy to be published and could be useful to share with the readers of the journal. The authors believe that the sharing of knowledge, experiences and new approaches should not be limited in dedicated journal but it should be disseminated as widely as possible. On the other hand, the manuscript is a report on the results obtained by the authors in the topic described in the title, we do not consider useful "explanatory figures" on operating principles of QqQ and/or IT, while we think that the introduction of our experiences could be a reason for further study by readers, perhaps by following the references provided.

The authors’ reply is clear and convincing. Indeed, the separation of isomers is a topic of high interest, and I may be underestimating the novelty provided here by applying an existing algorithm to a different MS analyzer. However, given (1) that this was how the manuscript was received and (2) given how significant the differences are as stated in the authors’ reply, additional figures explaining these differences (not merely visualizing the workings of QqQ/IT) may have been useful after all. In the end, it is an editorial decision whether a manuscript fits the scope of the journal and – if so – whether it is novel. I therefore refer this issue to the editor.

The authors believe they have already supported the differences and novelties in the manuscript. In the introduction section they report: "is it possible to transfer the MS/MS data post-processing LEDA approach to another type of tandem mass spectrometry system?". In our point of view this is a challenge and a novelty.

In the results section they report: "The application of the LEDA approach in the isomers recognition involves the MS/MS da-ta elaboration from a multistage MS system, such as triple quadrupole mass spectrometer (QqQ). However, also the ion trap (IT) MS analyzers can perform tandem MS analysis, although it manages the experiment differently. The IT operates the multistage MS experiments in the same site, using time to modify the conditions applied on studied ions [24]. Then, a sequence of time-dependent steps must be made to perform the selection of precursor ion, its fragmentation by CID mechanism and analysis of resulting product ions (Pis) [25-26]. For this reason, the MS/MS acquisition cycle on IT can be longer respect other multistage instruments, decreasing the frequency of sample data collection and influencing its chromatographic profile. Nevertheless, the MS/MS analysis from IT instruments shows some advantages: takes the advantages of Pis acquisition (complete MS/MS spec-trum), different excitation energy management and reiteration of the tandem MS experiment on a product ion (MSn) [27]”. We believe that in these sentences we have pointing out the differences between the mass analyzers and do not idea how to explain these with additional figures. To meet the reviewer's remark, we added 3 more refs that are fundamentals for this topic, useful for further study of the readers.

In previous revision, the reviewer’s remark: “A visualization of what SY, PiF, and PiY correspond to (in addition to including a verbal explanation near Fig. 2, since the mathematical explanations are currently provided in the much later section 4.4)”. But the figure 2 is an explaining figures about what SY, PiF, and PiY correspond to ..... However, all of section 2.1 is devoted to explaining these graphs and section 4.4 describes the used MS/MS parameters and how each data (SY, PiF, and PiY) are calculated. Also in this case, from our point of view, the planned and evaluation of the proposed study represents a novelty. Additional explaining figures are needed? In the manuscript were reported 8 figures, while in supplementary materials other 30 figures are showed. The authors believe that new figures will overload the manuscript without adding any information or novelty.
